# Lamin B1 mapping reveals the existence of dynamic and functional euchromatin lamin B1 domains

Laura Pascual-Reguant[1], Enrique Blanco [2], Silvia Galan [2,3], François Le Dily [2,4], Yasmina Cuartero[2,3], Gemma Serra-Bardenys[1,4], Valerio Di Carlo [2], Ane Iturbide[5], Joan Pau Cebrià-Costa[1], Lara Nonell[6], Antonio García de Herreros[4,7], Luciano Di Croce [2,8], Marc A. Marti-Renom [2,3,4,8] & Sandra Peiró[1]

Lamins (A/C and B) are major constituents of the nuclear lamina (NL). Structurally conserved lamina-associated domains (LADs) are formed by genomic regions that contact the NL. Lamins are also found in the nucleoplasm, with a yet unknown function. Here we map the genome-wide localization of lamin B1 in an euchromatin-enriched fraction of the mouse genome and follow its dynamics during the epithelial-to-mesenchymal transition (EMT). Lamin B1 associates with actively expressed and open euchromatin regions, forming dynamic euchromatin lamin B1-associated domains (eLADs) of about 0.3 Mb. Hi-C data link eLADs to the 3D organization of the mouse genome during EMT and correlate lamin B1 enrichment at topologically associating domain (TAD) borders with increased border strength. Having reduced levels of lamin B1 alters the EMT transcriptional signature and compromises the acquisition of mesenchymal traits. Thus, during EMT, the process of genome reorganization in mouse involves dynamic changes in eLADs.

[1] Vall d'Hebron Institute of Oncology, 08035 Barcelona, Spain. [2] Centre for Genomic Regulation (CRG), The Barcelona Institute of Science and Technology, Dr. Aiguader 88, Barcelona, Spain. [3] Structural Genomic Group, CNAG-CRG, Centre for Genomic Regulation (CRG), The Barcelona Institute of Science and Technology, Baldiri Reixac 4, Barcelona, Spain. [4] Departament de Ciències Experimentals i de la Salut, Universitat Pompeu Fabra (UPF), 08003 Barcelona, Spain. [5] Institute of Epigenetics and Stem Cells, D-81377 München, Germany. [6] Servei d'Anàlisi de Microarrays Institut Hospital del Mar d'Investigacions Mèdiques, Barcelona, Spain. [7] Programa de Recerca en Càncer, Institut Hospital del Mar d'Investigacions Mèdiques, Barcelona, Spain. [8] ICREA, Pg. Lluis Companys 23, Barcelona, Spain. Correspondence and requests for materials should be addressed to S.Pó. (email: speiro@vhio.net)

Nuclear genome folding occurs at multiple levels, and the dynamic folding of chromatin is known to be elemental in regulating gene expression. Alterations in these folding units are associated with multiple diseases and cancer[1]. One key level of organization involves the interaction between chromatin and the nuclear lamina (NL)[2,3]. Lamins (A/C and B) are type V intermediate filaments and are the major components of the NL. Chromatin regions that are in close contact with NL are called lamina-associated domains (LADs)[4–6]. These domains were initially identified using the DamID method, in which bacterial DNA adenine methyltransferase fused with lamin B1 methylate DNA regions that are in contact with NL[7]. LADs can be also detected by chromatin immunoprecipitation coupled with deep sequencing (ChIP-seq)[8–10] and by fluorescent in situ hybridization. LADs are formed by heterochromatin defined as chromatin regions with low gene frequency, transcriptionally silent, and enriched in the repressive histone marks, H3K9me2/3[11]. Importantly, LADs are extremely conserved between species, although some show a certain degree of dynamism[11]. Despite the extensive data published about NLs, little is known regarding its structural organization. High-resolution confocal microscopy and three-dimensional (3D) structured illumination microscopy showed that A- and B-type lamins form separated but interconnected meshworks with distinct roles[12,13]. Recently, it has been demonstrated that A- and B-type lamins assemble into tetrameric filaments of 3.5 nm, a structure surprisingly different than that of other cytoskeletal elements[14]. Moreover, these filaments are variable in length and are found to form both sparsely and densely packed regions, which are both detected around dense nuclear material that could be chromatin[14]. Unlike A-type, B-type lamins remain permanent farnesylated and carboxymethylated, and thus remain tightly associated with the membrane[15]. There is also evidence of the existence of a nucleoplasmic pool of lamins (A/C and B) that are assembled into stable structures with characteristics different from the A- and B-type lamins located in the NL[16]. This finding suggests that nucleoplasmic lamins may have a role distinct from that of perinuclear lamins[6,17]. In fact, recent ChIP-seq genome-wide studies have shown that lamin A/C contact euchromatin[18,19] and have suggested a functional role for lamin A/C in creating a permissive environment for gene regulation[18]. These findings are of high interest for two main reasons: (1) they demonstrate interactions between large euchromatin regions and nucleoplasmic lamin A; and (2) methodologically, they show how enrichment of different chromatin fractions can reveal distinct lamin A-associated domains[20]. Importantly, although DamID maps of lamin A and B are similar, a fraction of lamin A is found throughout the nucleus that is not detected by DamID, for yet unknown reasons[11,21]. This fact, together with evidence that lamins form separate but interconnected networks[12,13] and interact with nuclear structures distinct from the NL[6,16,17], led us to hypothesize that lamin B1 filaments could also interact with euchromatin.

Here we used euchromatin enrichment and ChIP-seq to map the localization of lamin B1, and we then analyzed its dynamism using the epithelial-to-mesenchymal transition (EMT) model[22]. The EMT program describes a series of events by which epithelial cells lose many of their epithelial characteristics and take on properties that are typical of mesenchymal cells. These cells undergo complex changes in both cell architecture and behavior[23]. Developmental biologists have long recognized that EMT is a crucial process for the generation of tissues and organs during embryogenesis of both vertebrates and invertebrates, and it also has an important role in pathological processes, such as fibrosis and cancer[24]. During progression to metastatic competence, carcinoma cells acquire mesenchymal gene expression patterns and properties, resulting in modified adhesive characteristics and in activation of proteolysis capacity and motility; these changes allow tumor cells to metastasize and establish secondary tumors at distant sites[22]. Recent studies suggest that chromatin re-organization during EMT is an essential step for the conversion of an epithelial cell into a mesenchymal cell[25,26]. However, local and 3D chromatin roles in EMT and tumorigenesis are incompletely understood.

Here we define the existence of LADs formed when lamin B1 contacts euchromatin regions, which we term euchromatin LADs (eLADs) to differentiate them from conventional LADs. Lamin B1 ChIP-seq from an enriched euchromatin fraction, RNA sequencing (RNA-seq), and assay for transposase-accessible chromatin using sequencing (ATAC-seq) allowed us to detect these domains and to analyze their dynamism in the context of EMT. In combination with whole-genome chromatin conformation capture (Hi-C), we demonstrated that eLADs are located in the active (A) compartment and that, at the onset of EMT, the amount of lamin B1 increased at TAD borders concomitantly with increased border strength. Moreover, over the time course, additional eLADs were formed involving genes that belong to the EMT pathway. Finally, depletion of lamin B1 from the euchromatin fraction massively affected the gene expression profile (as determined by RNA-seq) at a key time point of this cellular transformation, resulting in impaired EMT.

## Results

**Lamin B1 associates with euchromatin regions that are dynamic during EMT.** To induce EMT cellular transformation, normal mouse mammary epithelial cells (NMuMG) were treated with the transforming growth factor transforming growth factor (TGF)-β[27]. We first confirmed by confocal microscopy the presence of lamin B1 in the nuclear envelope as well as the nuclear interior of cells that were either untreated or treated with TGF-β for 8 or 24 h (Supplementary Fig. 1a). Colocalization with emerin, an integral protein of the NL[28], was only observed in the nuclear envelope (Supplementary Fig. 1b). We then performed ChIP-seq for lamin B1 on an enriched euchromatin fraction. As chromatin preparation is key to the identification of genomic-associated regions[18,29], chromatin was sheared by low sonication to obtain DNA fragments of 300–600 bp. This sonication condition favors shearing of accessible chromatin (e.g., euchromatin) but leaves heterochromatin regions intact[18,20,30]. In this way, heterochromatin regions can be excluded from Ilumina sequencing due to their size[18]; indeed, the bioanalyzer intensity profile showed a substantial fraction of fragments larger than 1 kb that were excluded from sequencing (Supplementary Fig. 1c). We identified significant lamin B1-positive sites (lamin B1 +) in chromatin in all three conditions (Fig. 1a), with a total of 4645 peaks in untreated cells, 10,484 peaks in 8 h TGF-β-treated cells, and 7083 peaks in 24 h TGF-β-treated cells. The distribution of ChIP-seq peaks across the genome showed a significant enrichment around the transcription start site (TSS) of specific genes (Fig. 1b) (of 4454 genes in untreated cells, 8755 genes in 8 h TGF-β-treated cells, and 6595 genes in 24 h TGF-β-treated cells) that did not overlap with canonical LADs (cLADs; Fig. 1a). Importantly, as all the immunoprecipitated chromatin could be analyzed by ChIP-quantitative PCR (qPCR) without size selection exclusion, we were also able to detect lamin B1 in a cLAD region (Fig. 1c). Moreover, the specificity of the antibody used for the genome-wide mapping of lamin B1 was confirmed by lamin B1 immunoprecipitation from NMuMG cells infected with lentivirus carrying either an irrelevant short hairpin RNA as a control (C), or specific for lamin B1 (knockdown, KD), in low-sonicated chromatin (Fig. 1d).

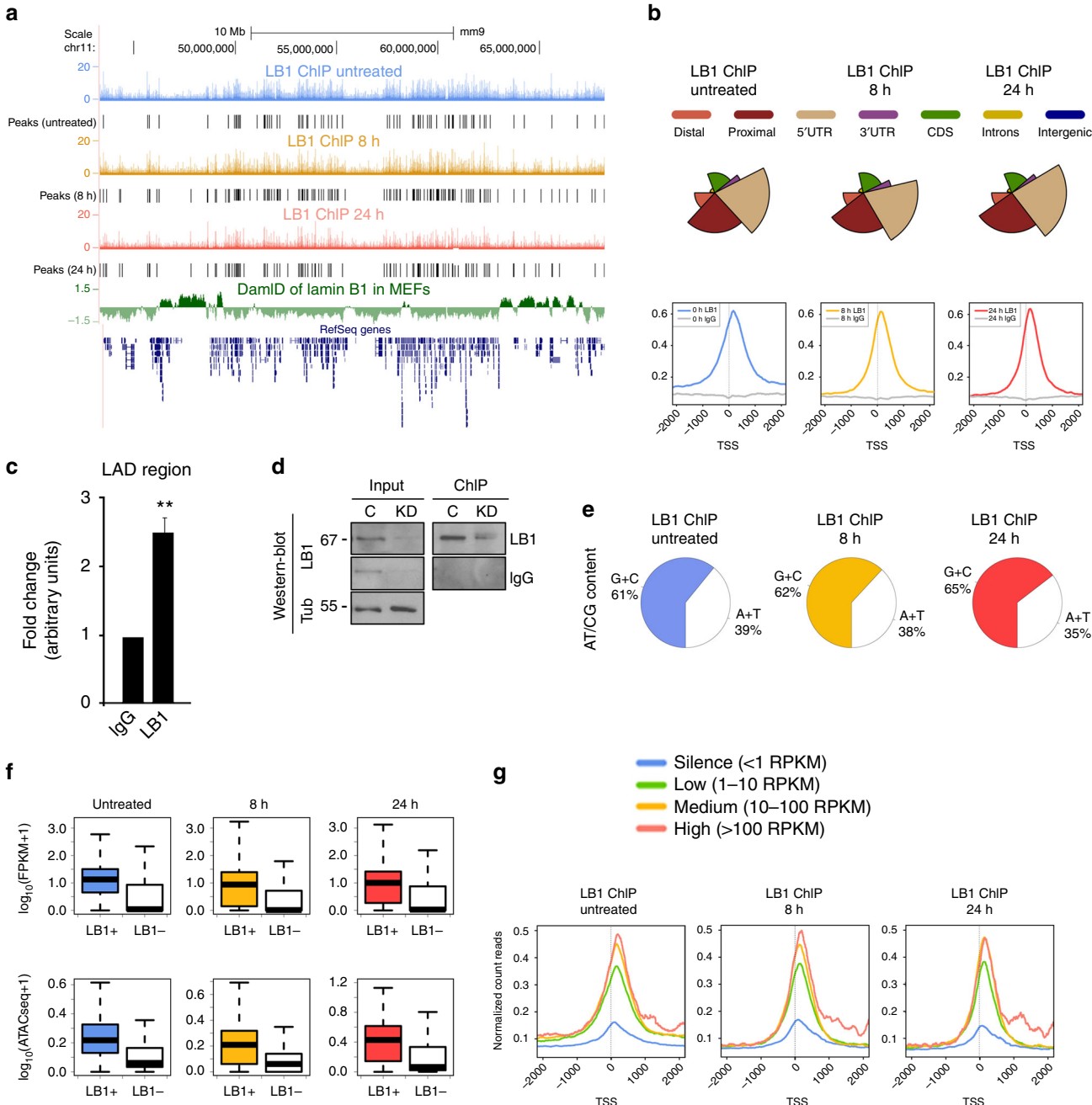

**Fig. 1** Lamin B1 associates with euchromatin regions. **a** UCSC Genome Browser overview of one region across chromosome 11 (mm9) of the lamin B1 ChIP-seq profiles in NMuMG cells that were untreated (blue) or treated with TGF-β for 8 h (orange) or 24 h (red). Lamin B1 + sites identified at each condition (black), previously published lamina-associated domains (LADs) (green), and the RefSeq gene track (dark blue) are shown. **b** Genome distribution of lamin B1 ChIP-seq peaks. The distal region is that within 2.5 and 0.5 Kb upstream of a gene's TSS, and the proximal region, within 0.5 Kb of a gene's TSS (top). The average distribution of lamin B1 ChIP-seq reads is shown, as well as the corresponding IgG control experiments, at 2 Kb around the TSS of lamin B1 + genes (bottom). **c** ChIP-qPCR of a canonical LAD (cLAD)-selected region. Data are expressed as the fold-change relative to data obtained from the IgG experiments. Data are shown as mean ± SD, $n = 4$. **d** Western blotting of chromatin immunoprecipitated with lamin B1 or IgG antibodies from normal or KD cells. **e** AT/CG content of the sequences of lamin B1 + sites. **f** Expression of lamin B1 + genes computed in FPKM (fragment per kilobase of transcript per million mapped reads) in untreated cells (blue) or those treated with TGF-β for 8 h (orange) or 24 h (red), as compared with the rest of genes in the genome (white) (top). Promoter ATAC-seq enrichment of lamin B1 + genes measured in number of normalized reads in untreated cells (blue) or those treated with TGF-β for 8 h (orange) or 24 h (red), as compared with the rest of genes in the genome (white) (bottom). The bottom and top fractions in the boxes represent the first and third quartiles, and the line, the median. Whiskers denote the interval between 1.5 times the interquartile range (IQR) and the median. **g** Average distribution of lamin B1 ChIP-seq reads at 2 Kb around the TSS of mouse genes that were classified into four categories (silent, low, medium, and high) according to their expression levels

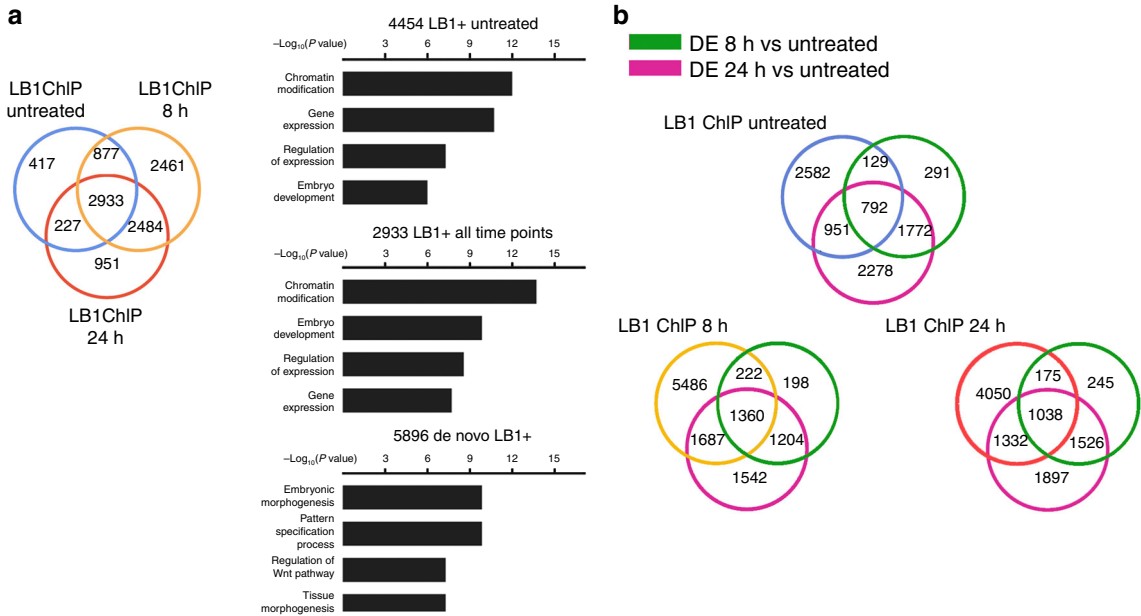

**Fig. 2** Lamin-positive sites are dynamic during EMT. **a** Venn diagram showing the overlap between lamin B1 + genes in untreated cells (blue) and cells treated with TGF-β for 8 h (orange) or 24 h (red) (left). Gene ontology (GO) terms of lamin B1 + genes only present in untreated conditions, present in all three time points, or newly formed at the onset of the EMT are depicted (right). **b** Venn diagram showing the overlap between lamin B1 + genes in untreated conditions (blue) or after treatment with TGF-β for 8 h (orange) or 24 h (red); genes differentially expressed (DE) at 8 h (green) and 24 h (pink) after TGF-β treatment are shown

Further characterization showed that, in contrast to A/T-rich constitutive LADs[31], lamin B1 + sites were strongly associated with C/G-rich sequences (Fig. 1e) and contained a substantially higher number of genes. To determine whether these genes were decorated with canonical euchromatin or were instead associated to heterochromatin histone marks, bioinformatics Enrichr analysis[32] was performed. We found how histone marks associated to these genes were characteristic for euchromatin (Supplementary Fig. 2a). We repeated the same analysis on genes that were not enriched in lamin B1 at each time point and H3K9me2/3, the classical heterochromatin histone mark, was enriched (Supplementary Fig. 2b). In addition, RNA-seq and ATAC-seq data from these samples showed that lamin B1-enriched genes were actively transcribed, with their promoters located in accessible chromatin regions (Fig. 1f). On the other hand, non-lamin B1-enriched genes presented low expression rates and less chromatin accessibility (white plots; Fig.1f). To analyze whether a correlation between lamin B1 binding and gene expression exists during EMT, we stratified the full set of mouse genes on each time point into four groups based on RNA-seq data (of silent, low, medium, and high expression). We then plotted the lamin B1 levels obtained by ChIP-seq for each gene set and observed a strong correlation between expression and lamin B1 around the TSS (the higher the levels of expression, the higher the lamin B1 enrichment) (Fig. 1g).

Comparing lamin B1 + sites during EMT revealed that, of the total of 10,350 target genes identified in the three time points, only 2933 genes (28%) maintained lamin B1 throughout the EMT process, whereas a vast majority (7417; 72%) changed at the onset of EMT (Fig. 2a). Taken together, these data suggest that lamin B1 can be found in expressed euchromatin regions associated with C/G regions that are gene-rich, accessible, decorated with euchromatin histone marks, and change dynamically during EMT transformation.

Gene ontology (GO) showed that genes enriched in lamin B1 only in untreated cells or that maintain their level of lamin B1 during the entire EMT process belong to chromatin modification and general gene transcription (Fig. 2a; for the full list of GO categories in each time point, see Supplementary Fig. 3a). We next focused on the set of genes that emerged as new lamin B1 targets upon EMT induction by TGF-β treatment (8 and 24 h). We found 5896 genes positive for lamin B1, which encoded proteins with functions belonging to embryonic morphogenesis and pattern specification, both of which strongly related to the EMT process (Fig. 2a). In order to identify potential drivers responsible for the increase in lamin B1 occupancy at these regions, we determined the enrichment of transcription factor (TF) motifs at these new lamin B1 sites. The enriched factors identified were involved in developmental and differentiation process (Supplementary Fig. 3b, c; Supplementary Table 1), and some of them are members of the TGF-β signaling pathway[33,34]. These data suggest that TGF-β-activated TFs may have a role in recruiting lamin B1 to these specific sites. Finally, the percentage of lamin B1 + genes for which we observed a change in expression at the onset of EMT were 42% (1872/4454) in the epithelial state, 37% (3269/8755) at 8 h after TGF-β treatment, and 38% (2545/6595) at 24 h after TGF-β treatment (Fig. 2b).

**Definition of eLADs and the putative role of lamin B1 as an architectural protein.** cLADs have been traditionally highlighted as genomic regions enriched in lamins that emerged from comparison with the corresponding control experiments[4,18]. We similarly defined eLADs as clusters of neighboring lamin B1 + sites that occur within a delimited region of the genome (see Supplementary Methods). We identified 2051 eLADs in untreated cells, 2429 eLADs in 8 h-treated cells, and 2949 eLADs in 24 h-treated cells (Fig. 3a), with an average size of 0.34 Mb (and therefore smaller than LADs, which are about 1 Mb[4]). The coverage of the mouse genome for each set of eLADs was 25.9%, 31.7%, and 40.1%, respectively. As expected, genes within eLADs were active and located in accessible chromatin (Fig. 3b).

Given that lamin A/C also occupies euchromatin regions[18], we analyzed the degree of overlap between both sets of ChIP-seq

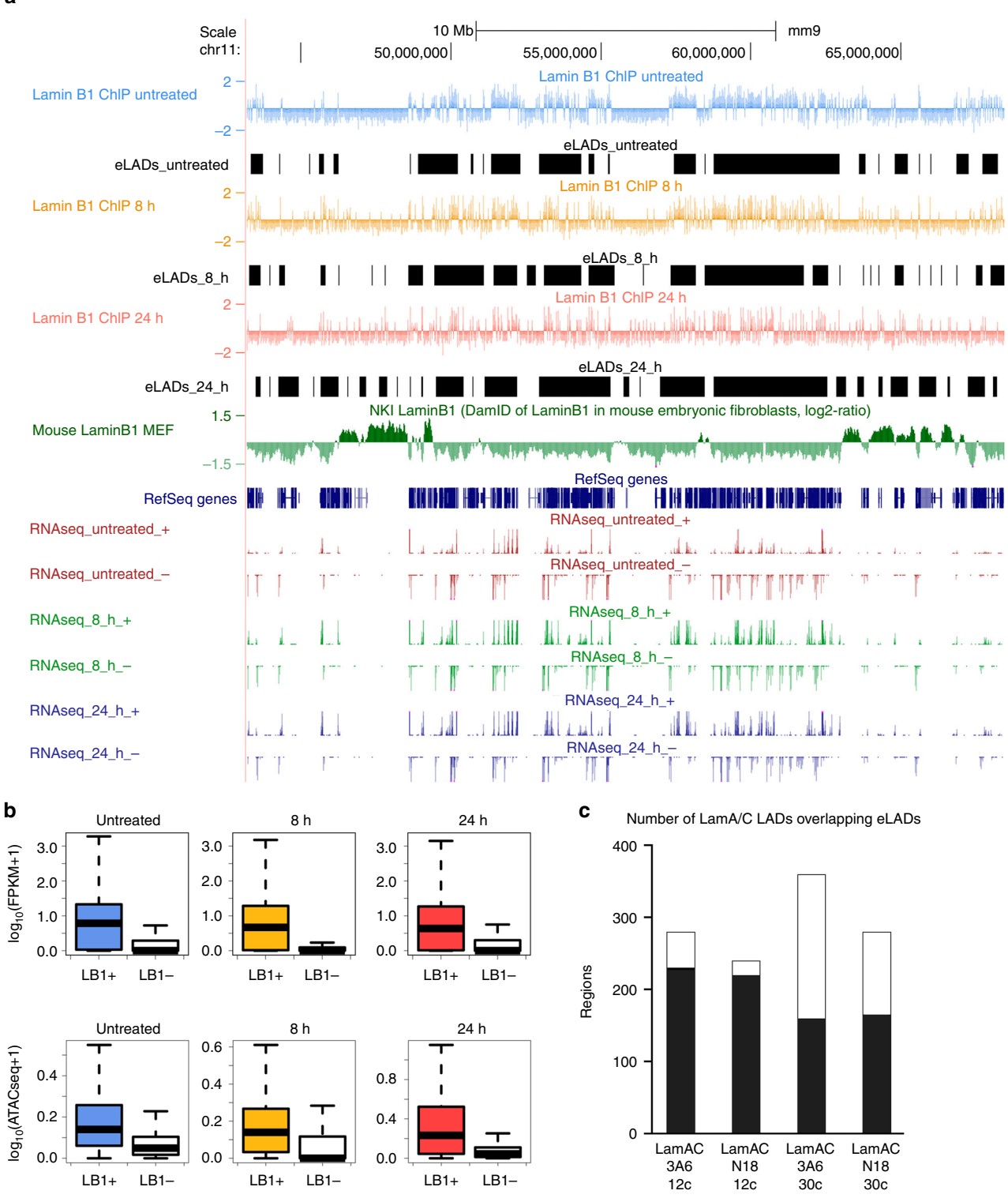

**b**

Untreated / 8 h / 24 h — log$_{10}$(FPKM+1) boxplots for LB1+ and LB1−

Untreated / 8 h / 24 h — log$_{10}$(ATACseq+1) boxplots for LB1+ and LB1−

**c** Number of LamA/C LADs overlapping eLADs

Regions plotted for LamAC 3A6 12c, LamAC N18 12c, LamAC 3A6 30c, LamAC N18 30c

data. As expected, eLADs had an extended overlap with euchromatin regions enriched in lamin A/C (Fig. 3c).

Intermediate filaments are major contributors to cell architecture[35]. As genome function relies on the genomic spatial architecture[36], we hypothesized that lamin B1 could help to shape the new mesenchymal genome architecture. Genome organization and architecture was thus analyzed by Hi-C experiments[37] during EMT transformation. Sequencing Hi-C libraries from untreated cells or those treated with TGF-β (for 8 or 24 h)

resulted in 89–106 M valid interactions per time point (Supplementary Table 2), which allowed us to generate chromosome-wide interaction maps at 100 and 40 Kb resolution for A/B (activated/repressed) compartments and TAD analyses, respectively. Hi-C maps showed overall compartment changes between untreated and 8 h-treated cells, with about 2400 bins of 100 Kb that changed compartments (Fig. 3a). These structural changes were further reinforced after a 24 h TGF-β treatment, with 2228 bins of EMT-related genes changing from B to A and 942

**Fig. 3** Definition of eLADs. **a** UCSC Genome Browser overview of one region across the chromosome 11 (mm9) containing the following information (from top to bottom): lamin B1 ChIP-seq profiles subtracting the IgG control in untreated NMuMG cells (blue) or in cells treated with TGF-β for 8 h (orange) or 24 h (red); eLADs identified in each condition (black); previously published LADs from mouse embryonic fibroblast cells (green); and RNA-seq strand-specific expression profiles at each time point (red for untreated NMuMG cells, green for cells treated with TGF-β for 8 h, and blue for cells treated for 24 h). **b** Gene expression within eLADs (given in FPKM) in NMuMG cells that were untreated (blue) or treated with TGF-β for 8 h (orange) or 24 h (red), as compared with the rest of genes (white) (left). Promoter ATAC-seq enrichment of lamin B1+ genes (measured in number of normalized reads) in NMuMG cells that were untreated (blue) or treated with TGF-β for 8 h (orange) or 24 h (red), as compared with the rest of genes (white) (right). Genes from eLADs were more likely to be expressed and located in accessible chromatin than other genes. The bottom and top fractions in the boxes represent the first and third quartiles, and the line, the median. Whiskers denote the interval between 1.5 times the interquartile range (IQR) and the median. **c** Bar plot showing the overlap between eLADs and lamin A/C LADs obtained after low or high sonication (12 cycles or 30 cycles, respectively) and with two different antibodies (3A6 and N18)

changing from A to B (Fig. 4a, Supplementary Fig. 4). We next sought to analyze to which extent the dynamic changes of lamin B1 + sites and eLADs were related to genome architecture (i.e., in A/B compartments and at TAD borders). As expected, transcription, ATAC signal, lamin B1 + sites, and eLADs were enriched in A compartment (Fig. 3b), whereas cLADs were located in B compartments. In addition, we observed that lamin B1 + sites and eLADs decreased in A compartments and increased in B compartments during EMT (Fig. 4b); these corresponded to newly formed eLADs (Supplementary Table 3).

As borders between TADs are enriched in TSS, located in transcriptional active genomic regions, and enriched in architectural proteins[38–40], we analyzed TAD border behavior at the onset of the EMT and their relation to lamin B1. We observed that, although TAD borders were conserved during EMT transformation (with 50% of all borders conserved over the three time points), they increased in overall strength with longer treatment, indicative of more stable borders and increased intra-TAD interactions (Fig. 4c, left panel); this may reflect novel and biologically relevant chromatin interactions within TADs[36,41]. As changes in border strength could be due to changes in gene transcription[41], we checked the transcription rates after treatment at the TAD borders that had increased levels of lamin B1. Intriguingly, transcription was maintained in these TAD borders during EMT, suggesting that the increase in border strength is not due to changes in transcription (Fig. 4c, right panel). The increased number of lamin B1 + sites after TGF-β treatment (Fig. 2a) correlated with an increase in the percentage of lamin B1-containing borders, enhanced TAD border strength, and an enrichment of lamin B1 at TAD borders (with p-values of $10^{-28}$, $10^{-68}$, and $10^{-32}$ in untreated, 8 h-treated, and in 24 h-treated cells, respectively) (Fig. 4d). Overall, these results suggested that lamin B1 could contribute to 3D genome organization during EMT.

**Altered lamin B1 levels impair EMT.** Finally, we assessed the functional relevance of lamin B1 in a KD model, using NMuMG cells. It is noteworthy that the KD was kept at about 50% efficiency to avoid indirect effects (Fig. 5a). Given the low protein turnover of nuclear lamins once these are stably integrated in the NL[42], we reasoned that knocking down lamin B1 would mainly affect non- or less-NL-integrated lamin B1. Nuclei from control and KD NMuMG cells were extracted, and soluble and loosely bound chromatin proteins were separated from the insoluble chromatin fraction. Importantly, the soluble fraction of lamin B1 was affected to a greater extent in the KD cells as compared with control cells (Fig. 5b). To further confirm that the KD affected mainly the nucleoplasmic (or less-NL-integrated) lamin B1 fraction rather than the high-NL fraction, we used fluorescence recovery after photobleaching (FRAP). NMuMG cells transfected with human mCerulean-lamin B1 were subjected to short hairpin RNA (shRNA) lentivirus infection (with a control or human

lamin B1-specific shRNA). Although recovery from bleaching was similar for the NL mCerulean-lamin B1 in both control and KD conditions, the nucleoplasmic fraction of mCerulean-lamin B1 recovered faster in the control cells than in the KD cells (Fig. 5c, left panel). Quantification data showed the mean recoveries for ten cells (Fig. 5c, right panel). Indeed, lamin B1 occupancy was reduced by 50% in lamin B1 + sites in KD cells, as shown by ChIP-qPCR (Fig. 4d), whereas cLADs were not affected (Fig. 5e). Under these conditions, KD cells had the same growth rate as control cells (Supplementary Fig. 5a) did not have a disrupted NL (Supplementary Fig. 5b) and did not enter into senescence (Supplementary Fig. 5c), all traits related with a massive loss of lamin B1[43].

Next, we analyzed the transcription profile of lamin B1 KD NMuMG cells treated with TGF-β during EMT by RNA-seq at each time point (Fig. 6). Principle component analysis revealed that KD cells at 8 h TGF-β were the most highly divergent from the control cells (Fig. 6a). Importantly, this is the time point in where we observed the highest number of newly formed lamin B1 + regions. Further, differentially expressed genes showed again a maximum dependence of lamin B1 at the 8 h TGF-β time point, with more than 2000 genes differentially expressed as compared with control cells (Fig. 6b). Strikingly, around 50% of these genes were direct lamin B1 targets (bar colors, Fig. 6b). Moreover, the differentially expressed genes that were upregulated in KD conditions did not belong to the EMT pathway, which completely changed the EMT transcriptional program (Fig. 6c, Supplementary Fig. 6). Indeed, western blot data showed alterations in classical EMT markers, a lack of fibronectin, N-cadherin upregulation, E-cadherin downregulation, and a loss of migrative and invasive capacities normally acquired during EMT (Fig. 6d, e). Finally, ChIP-qPCR in LB1 + EMT genes at 8 h TGF-β (such as fibronectin, vimentin, and twist) showed loss of lamin B1 enrichment in KD conditions as compared with control cells (Fig. 6f). Importantly, overexpressing human GFP-lamin B1 in KD cells restored the migratory capacity of mesenchymal cells (Fig. 6g).

## Discussion

The presence of lamins in the nuclear interior has been long known but considered to be a transient pool on the way to assembly into the NL. Although B-type lamins remain permanently farnesylated and carboxymethylated (and therefore tightly associated to membranes[15]), they have also been reported to localize in the nuclear interior[44]. Polymerized lamins A and B exists within the nucleoplasm[6,16,17,45], but their specific organization, state of polymerization, solubility, and degree of integrity within the NL remains to be determined. Lamins are known to bind DNA, histones, and histone-binding proteins[46,47], and recent reports have also shown that lamin A/C bind to euchromatin regions, with an important role in gene regulation[18]. It is thus conceivable that all type of lamins in the

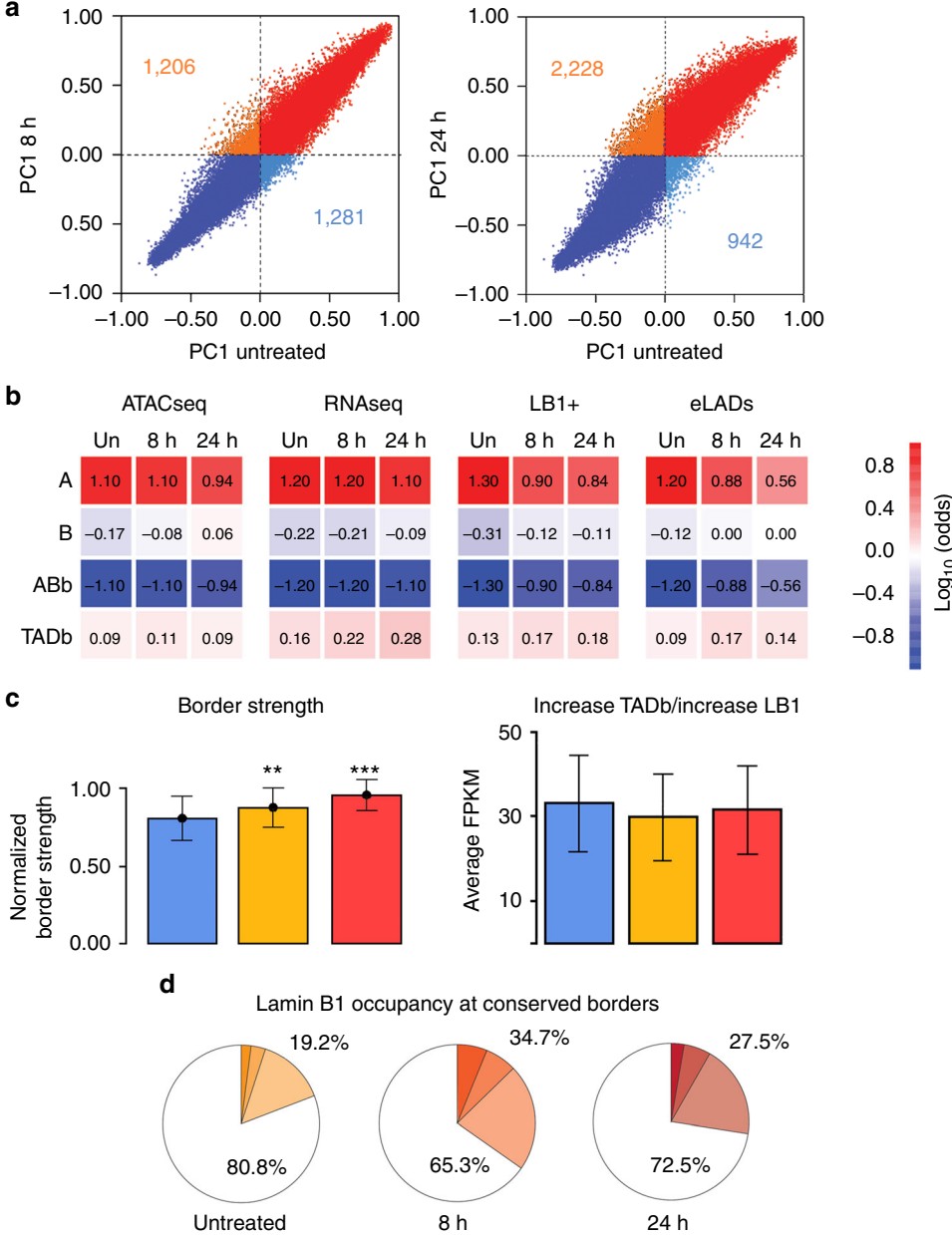

**Fig. 4** Putative role of lamin B1 as an architectural protein. **a** A/B compartment dynamics. Scatterplots of PC1 values indicate compartment changes between untreated cells and cells treated with TGF-β for 8 or 24 h. Dots indicate 100 Kb bins that changed compartment with respect to untreated cells (from B to A, in orange; from A to B, in cyan). Numbers indicate the total bins that changed compartment. **b** Contingency table analysis for enriched and depleted features according to distinct genome architectural landmarks. Red to blue color indicate positive to negative log odds of the feature in genomic bins assigned to A or B compartment, or TAD borders. All log odds are statistically different than zero ($p < 0.05$, Fisher's exact $t$-test). **c** TAD border dynamics. Bar plot of normalized border strength for conserved borders in untreated cells (blue) or those treated with TGF-β for 8 h (orange) or 24 h (red) (left). Expression of genes within TAD borders in which lamin B1 increased (computed in FPKM) in untreated cells or those treated with TGF-β for 8 or 24 h (right). **d** Border occupancy by lamin B1 in conserved TADs. Pie charts of 1, 2, or $\geq 3$ called peaks (dark to light color) of lamin B1 ChIP-seq in a conserved border

nuclear interior contribute to chromatin organization, although little is known about their potential roles in gene regulation. Lamins A/C have a functional role accommodating chromatin environment for gene regulation[18]. We now present evidence that the presence of lamin B1 in euchromatin is dynamic and has a role in the execution of the EMT transcriptional program, suggesting that lamin B1 also has a crucial role in gene regulation. We believe that this role is not exclusive to EMT. In fact, the GO categories found in genes enriched in lamin B1 in the epithelial state belong to general transcription, which suggests a general role of lamin B1 in gene regulation in response to a given stimulus.

Given that intermediate filaments are the major contributors of cell architecture[35] and, specifically, that nuclear lamins provide structural stability to the nucleus[48,49] and participate in multiple nuclear activities, we propose a role for lamin B1 in helping the 3D chromatin rearrangements occurring during EMT (Fig. 7). Hi-C data in combination with lamin B1 ChIP-seq showed that eLADs are present in the A compartment during the entire process, and that there is a significant increase of eLADs in the B

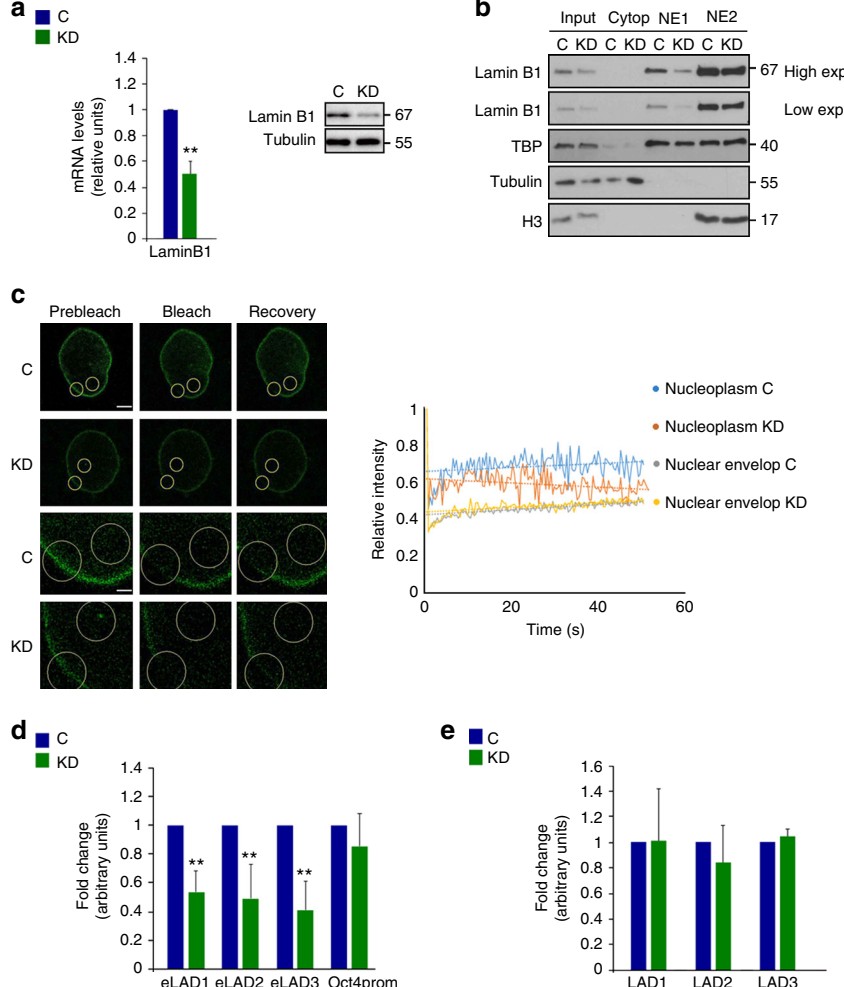

**Fig. 5** Knockdown of lamin B1 mainly affects non-NL integrated lamin B1. **a** qRT-PCR showing mRNA levels of lamin B1 in KD NMuMG cells. Expression levels were normalized to an endogenous control and expressed relative to the control-infected cells (C), which was set as 1. Data are presented as mean ± SD, $n = 5$. Left panel, representative western blotting for LB1 and tubulin (as a loading control) from C and KD NMuMG cells. Images are representative of three independent replicates (right). **b** Fractionation of nuclei from C and KD NMuMG cells to obtain cytoplasmic proteins, soluble proteins (NE1), and insoluble proteins (NE2), as monitored by western blotting using TBP as a NE1 control, tubulin as a cytoplasmic control, and H3 as a NE2 control. Images are representative of two independent replicates and five technical replicates. **c** Confocal fluorescence recovery after bleaching (FRAP) in C and KD cells expressing lamin B1-mCerulean. Circles denote the areas of bleaching. Each column displays prebleach, bleach, and recovery images. Rows 3 and 4 are magnifications of rows 1 and 2, respectively. Scale bar: 5 µm (inset 1 µm) (left). Images are representative of ten cells for each condition. Graphic representation of the relative intensity after 60 s of nuclear envelope and nucleoplasm recovery from bleaching in C and KD cells (right). Data are shown as mean, $n = 6$. **d** ChIP-qPCR of three selected lamin B1 + regions and a negative control (Oct4prom) selected from a negative lamin B1 + region, in C and KD NMuMG cells. Data from real-time PCR (qPCR) amplifications were normalized to the input and expressed as the fold-change relative to data obtained in C condition, which was set as 1. Data are shown as mean ± SD, $n = 5$. **e** ChIP-qPCR of three cLAD-selected regions in C and KD NMuMG cells. Data from qPCR amplifications were normalized to the input and expressed as the fold-change relative to data obtained in C condition, which was set as 1. Data presented as mean ± SD, $n = 3$

compartment over time during EMT. Data analyses showed that this increase in the B compartment corresponds to newly formed eLADs, rather than simply reflecting a movement of eLADs from A to B. Interestingly, genes present in these new eLADs have to be in a repressive state when cells become mesenchymal, further supporting this function as a chromatin organizer. Moreover, the correlation of lamin B1 enrichment and border strength suggests a possible role of lamin B1 as an architectural protein that has a critical role in the establishment of a new genomic architecture during EMT (Fig. 7). The increase in border strength might be a consequence of transcriptional changes that occur during EMT. However, we did not observe dramatic transcriptional changes at TAD borders enriched in lamin B1, suggesting that transcription is not the main cause of these changes. Our results are in

accordance with other models that have shown that redistribution of architectural proteins is responsible for establishing new genome interactions[41]. We cannot rule out that other proteins are also involved in establishing long-range interactions. Notably, RNA polymerase II and/or its associated—but not the process of transcription per se—have been identified as mediators of interactions throughout the A compartment[50,51].

Hundreds of proteins are able to interact with lamins[52]. As different tissues express different sets of lamina-associated proteins[53], the organization of lamina filaments can potentially vary significantly among tissues. Our results show that, once the EMT program is activated, new lamin B1 sites are formed that are enriched in TF-binding motifs belonging to developmental programs, including the TGF-β signaling pathway, suggesting a

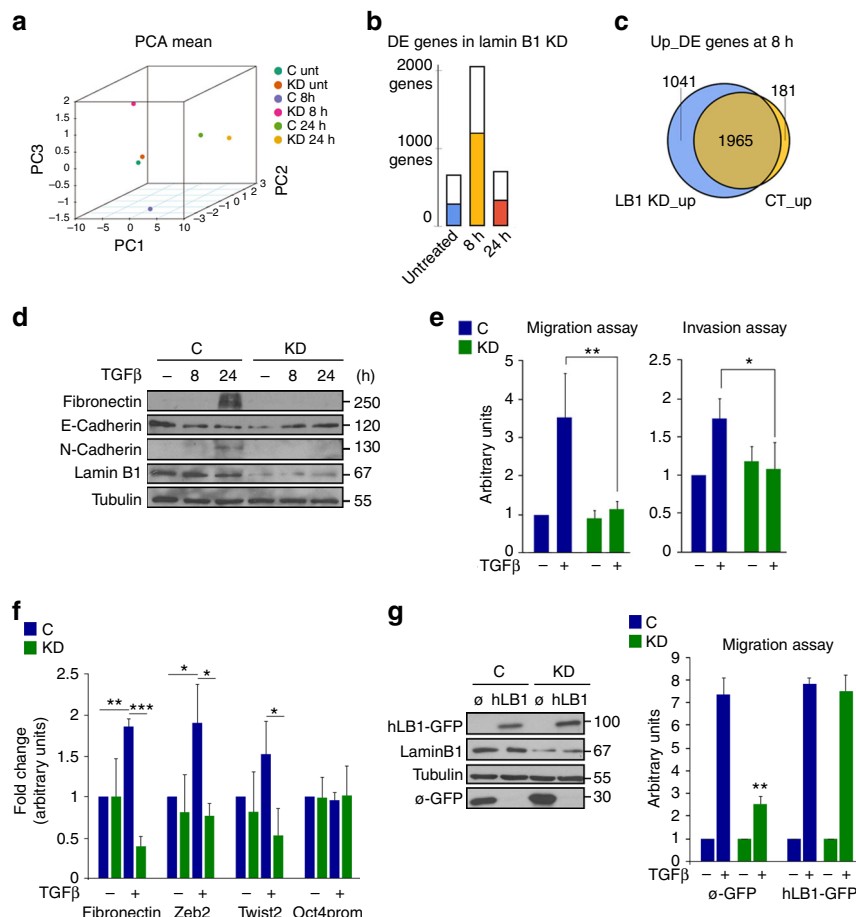

**Fig. 6** Lamin B1 is essential for EMT. **a** Principle component analysis (PCA) plot of RNA-seq based expression profiling in KD conditions. Each time point was normalized to its respective control. **b** Number of genes that were differentially expressed (DE) upon lamin B1 knockdown (KD) during the EMT process. Inside each bar, the proportion of genes that are direct targets of lamin B1 in the ChIP-seq experiment of NMuMG cells untreated (blue) or treated with TGF-β for 8 h (orange) or 24 h (red) are shown. **c** Venn diagram representing the overlap between upregulated genes after 8 h of TGF-β treatment in KD and C conditions. **d** Western blotting showing protein levels of different EMT markers and lamin B1 in C and KD NMuMG cells that were untreated or treated with TGF-β for 8 or 24 h. Tubulin was used as a loading control. Images are representative of three independent replicates. **e** Migration (left panel) and invasion (right panel) assays performed with C and KD NMuMG cells after 24 h of TGF-β treatment. Data are presented as mean ± SD, n = 3. **f** Chip-qPCR of three selected EMT-related lamin B1 + regions at 8 h after TGF-β treatment in C and KD conditions. One of the selected regions from Fig. 4d was used as a negative control (prOCT4). Data from qPCR were normalized to the input and expressed as the fold-change relative to the untreated C condition, which was set as 1. Data are shown as mean ± SD; n = 3. **g** Western blotting of overexpressed GFP-control or GFP-hLB (human LB1) from C or KD NMuMG cells. Tubulin was used as a loading control (left panel). Images are representative of three independent replicates. Migration assay of C and KD NMuMG cells transfected with either GFP-control or GFP-hLB (right panel). Data are shown as mean ± SD; n = 2

putative role for these TFs in recruiting lamin B1 (although further experiments will be required to demonstrate this). The apparently distinct configuration between epithelial and mesenchymal cell types, together with the fact that different antibodies were used, might explain why Gesson et al.[18] did not detect lamin B1 in euchromatin regions, although they were also working with low sonication conditions.

Finally, it is still unclear why the presence of lamins (A/C and B1) in contact with euchromatin regions cannot be detected by the DamID method. Notably, DamID was recently proposed to favor the identification of stable interactions[21]. DamID methylation is modulated by the local chromatin structure, which means that open chromatin regions (euchromatin) may reach similar or even higher levels than the sites of specific methylation. Correction of the methylation levels by Dam unfused proteins could be responsible for a loss of lamin-euchromatin signal, and we now know that there is a more stable lamin pool in the NL that will always provide stronger signal.

To summarize, although lamin B1 filaments in contact with repressed chromatin are mainly located in the B compartment, where they form cLADs[4], our present work also shows that transcriptionally active chromatin in the A compartment can interact with lamin B1, to form eLADs. This is consistent with previous models showing lamin A/C also contact euchromatin[18], and with the fact that these two filaments form an interconnected meshwork[12,13]. Importantly, these two types of lamin B1 domains —cLADs and eLADs—behave differently: while cLADs are static, eLADs are dynamic and change during EMT. Moreover, the correlation of lamin B1 enrichment and border strength suggests a possible participation of these domains in establishing new genomic architecture during EMT (Fig. 7). Many questions still remain to be addressed: is the molecular structure of lamin B filament networks distinct in cLADs and eLADs? Is this lamin B1 pool soluble, or is it still membrane-bound but less stably integrated (and therefore more dynamic)? Are eLADs universal? Do they have a role in all types of cellular transformation processes?

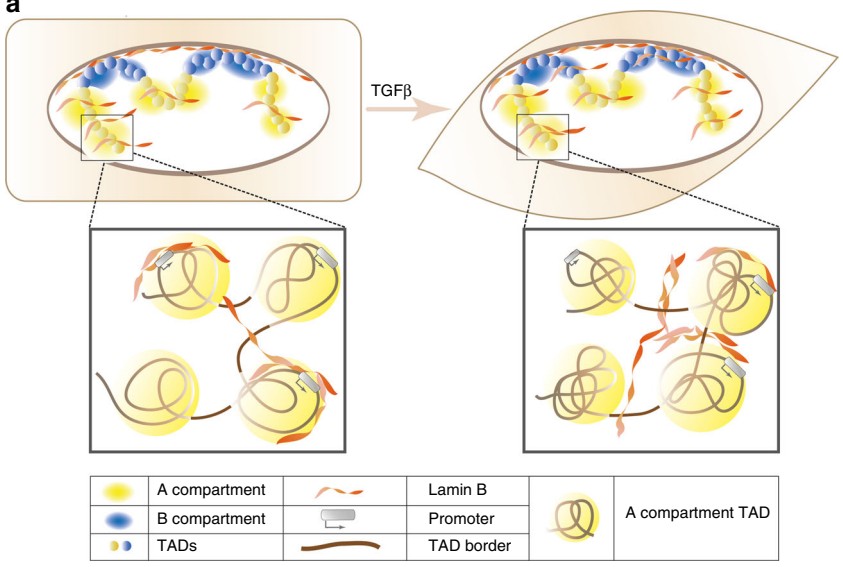

**Fig. 7** Schematic representation of the dynamism of eLADs during the EMT process

Which are the signals and/or TFs responsible for recruiting lamin B1 to different loci? Are the changes in interactions the cause or consequence of lamin B1 enrichment at TAD borders? Answers to these questions will contribute to our understanding of how genome is reorganized during cellular transformation, and the role of euchromatin-lamin contacts in this re-organization and, therefore, in gene regulation.

## Methods

**Nuclear fractionation**. Nuclear fractionation of infected NMuMG cells with either irrelevant shRNA or LB1 shRNA was performed using a Nuclear Extract Kit (ab219177, Abcam).

NMuMG cells were seeded in p150 plates and the assay was performed at 4 °C following manufacturer's instructions.

**Cytochemical staining for SA-β-gal**. Cytochemical staining for senescence-associated galactosidase (SA-β-gal) was performed using a Senescence β-Galactosidase Staining Kit (Cell Signaling Technology) at pH 6.0.

NMuMG cells infected and selected with puromycin were seeded in six-well plates, and 48 h after the assay was performed following manufacturer's instructions. In brief, culture medium was removed from the plates and two washes with phosphate-buffered saline (PBS) were performed. Cells were then fixed with Fixative solution provided with the Kit during 15 min at room temperature. Two more washes with PBS were performed before the addition of β-Galactosidase staining solution. Plates were sealed with parafilm and incubated overnight at 37 °C in a dry incubator. To avoid evaporation, corners of the plates were filled with PBS.

**ChIP experiments**. ChIP experiments were performed as described[54]. In brief, NMuMG cells were crosslinked in 1% formaldehyde for 5–10 min at 37 °C. Crosslinking was stopped by adding glycine to a final concentration of 0.125 M for 2 min at room temperature. For nuclear fractions, cells were scraped with cold soft-lysis buffer (50 mM Tris-HCl, 10 mM EDTA, 0.1% NP-40, and 10% glycerol) supplemented with protease inhibitors. Samples were then centrifuged at $800 \times g$ for 15 min and the nuclei pellets were lysed with SDS-lysis buffer (1% SDS, 10 mM EDTA, and 50 mM Tris pH 8) supplemented with protease inhibitors. Extracts were sonicated to generate 200–600 bp DNA fragments, incubated on ice for 20 min, centrifuged at $16,000 \times g$ for 10 min, and then diluted 1:10 with dilution buffer (0.01% SDS, 1.1% Triton X-100, 1.2 mM EDTA, 16.7 mM Tris pH 8 and 167 mM NaCl). For ChIP analysis, the primary antibody or an irrelevant antibody (IgG) was added to the sample, and the mixture was incubated overnight with rotation at 4 °C. Chromatin bound to the antibody was then immunoprecipitated using unblocked protein A beads (Diagenode) for 3 h with rotation at 4 °C. Precipitated samples were then washed three times with low-salt buffer (0.1% SDS, 1% Triton X-100, 2 mM EDTA, 20 mM Tris-HCl pH 8.0, 150 mM NaCl) and with high-salt buffer (0.1% SDS, 1% Triton X-100, 2 mM EDTA, 20 mM Tris-HCl pH 8.0, 500 mM NaCl), and twice with LiCl buffer (250 mM LiCl, 1% Nonidet P-40, 1% sodium deoxycholate, 1 mM EDTA, 10 mM Tris-HCl pH 8.0) using columns. To verify antibody specificity, samples were eluted with 2 × western blotting loading buffer for 5 min at 95 °C. Proteins were separated by SDS–polyacrylamide

gel electrophoresis and probed with the anti-lamin B1 antibody. For qPCR detection of genomic regions of ChIP-sequencing analysis, washed samples were treated with elution buffer (100 mM $Na_2CO_3$ and 1% SDS) for 1 h at 37 °C and then incubated at 65 °C overnight with the addition of a final concentration of 200 mM NaCl to reverse the formaldehyde crosslinking. After proteinase K solution (0.4 mg/ml proteinase K (Roche), 50 mM EDTA, 200 mM Tris-HCl pH 6.5) treatment for 1 h at 55 °C, DNA was purified with MinElute PCR purification kit (Qiagen) and eluted in nuclease-free water. Genomic regions were detected by qPCR. Primers used are listed in Supplementary Table 4 (Supplementary Table 4). Results were quantified relative to the input and the amount of irrelevant IgG immunoprecipitated in each condition.

For ChIP-seq analysis, two parallel ChIPs were performed and mixed after elution with nuclease-free water. The NEBNext Ultra DNA library Prep Kit for Illumina was used to prepare the libraries and samples were sequenced using Illumina HiSeq 2500 system.

**MTT assays**. Cells previously infected and selected for 48 h with puromycin were counted with Neubauer's Chamber and 10,000 cells/condition were seeded in 96-well plates by triplicate.

MTT (3-(4,5-dimethylthiazol-2-yl)-2,5-diphenyltetrazolium bromide) assays were performed by adding 0.5 mg of MTT (Sigma) per mL of Dulbecco's modified Eagle's medium (DMEM) without fetal bovine serum (FBS) for 3 h at 37 °C to determine the percentage of viable cells. Cells were solubilized with dimethyl sulfoxide-isopropanol (1:4). The absorbance of insoluble formazan (purple) at 590 nm, which is proportional to the number of viable cells, was then determined. Cell viability was quantified during four consecutive days.

**Migration and invasion**. For migration experiments, control (Irrelevant shRNA) and KD (shRNA_LB1) NMuMG cells were treated with TGF-β. After 24 h, 50,000 cells were resuspended in DMEM 0.1% FBS–0.1% bovine serum albumin, reseeded on a transwell filter chamber (Costar 3422) and incubated for 6–8 h. For invasion assays, cells were placed in Matrigel-coated transwell filter (BD356234) and incubated for 12–16 h. In both cases, DMEM with 10% FBS was added to the lower chamber and used as a chemoattractant. Non-migrating and non-invading cells were removed from the upper surface of the membrane, whereas cells that adhered to the lower surface were fixed with paraformaldehyde 4% for 15 min. Nuclei were stained with PBS-DAPI (4′,6-diamidino-2-phenylindole) (0.25 μg/mL). DAPI-stained nuclei were counted in four fields per filter by ImageJ software.

**Fluorescence recovery after photobleaching**. NMuMG cells were first transfected with human mCerulean-Lamin B1-10, which was a gift from Michael Davidson (Addgene plasmid #55380)[55] using *Trans*IT-X2® Dynamic Delivery System (Mirus Bio LLC). After 8 h, transfected cells were infected using irrelevant shRNA and human shRNA LB1 and selected using puromycin (1 μg/mL) for 24 h. Transfected and infected cells were seeded in 35 mm MatTek dishes. After 24 h, Leica TCS SP5 confocal system with the LAS-AF application wizard was used to perform FRAP experiment. Cells were kept in a fully incubated ($CO_2$ and 37 °C) chamber, while imaging with a ×63, 1.4 objective and 458 nm laser line of the Argon laser for excitation. Selected nuclear areas were bleached for five times using maximum laser intensity and 100 frames after the photobleach were collected, with 370 ms intervals, using the minimal laser power required. Before photobleaching,

10 frames were collected as an internal control of the experiment. Signal recovery was measured by ImageJ software using pre-bleached (pb), background (bg), and non-bleached (nb) areas (regions of interest, ROIs) to normalize the data. For every time point, the data was normalized according to the formula: $(ROI_b - ROI_{bg})/(ROI_{nb} - ROI_{bg}) / (pbRO_{Ib} - pbROI_{bg})/(pbROI_{nb} - (pbROI_{bg})$[56].

**Statistical analysis**. Statistical significance was assessed using a two-tailed unpaired Student's $t$-test. The symbols *, ** and *** indicates significant differences with $p < 0.05$, $p < 0.01$ and $p < 0.001$, respectively.

**RNA sequencing**. RNA-seq experiments were performed with two biological replicates of NMuMG cells (with or without shRNA of either control or KD) that were untreated or treated with TGF-β for 8 or 24 h. After RNA extraction with GenElute$^{TM}$ Mammalian Total RNA Miniprep Kit (Sigma-Aldrich), samples were sequenced using the Illumina HiSeq 2500 system.

**ATAC sequencing**. The ATAC experiment was performed as described[57]. NMuMG cells were either untreated or treated with TGF-β for 8 or 24 h, and then collected and treated with transposase Tn5 (Nextera DNA Library Preparation Kit, Illumina). DNA was purified using MinElute PCR Purification Kit (Qiagen). All samples were then amplified by PCR using NEBNextHigh-Fidelity 2× PCR Master Mix (New Englands Labs) with primers containing a barcode to generate libraries. DNA was again purified using MinElute PCR Purification kit and samples were sequenced using Illumina HiSeq 2500 system.

**Hi-C experiments**. Hi-C libraries were generated from NMuMG cells (that were untreated or treated with TGF-β for 8 or 24 h) according the previously published Hi-C protocol, with minor adaptations[58]. Five million cells were crosslinked with 1% formaldehyde for 10 min at room temperature. Before permeabilization, cells were treated for 5 min with trypsin to obtain single cells. DNA was digested with 400 units of Dpn II and the ends of restriction fragments were labeled using biotinylated nucleotides and ligated in a small volume (in situ Hi-C). Libraries were generated independently in the three conditions (e.g., treated with TGF-β for 8 or 24 h, or untreated), controlled for quality, and sequenced on an Illumina HiSeq 2000 sequencer.

**Data availability**. Sequencing samples (raw data and processed files) are available at NCBI GEO under the accession code GSE96033.

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

## Acknowledgements

We thank J. Querol for technical assistance and V.A. Raker for manuscript editing. This work was supported by grants from the Instituto de Salud Carlos III (ISCIII) FIS/FEDER (PI15/00396; CPII14/0006), Ministerio de Economía y Competitividad (MINECO) (SAF2013-40922-R1; FPU14/0407; BFU2016-75008-P), Agencia Estatal de Investigación (AEI) and Fondo Europeo de Desarrollo Regional-FEDER (SAF2016-76461-R), Generalitat de Catalunya (2014 SGR 32), Fundació FERO, Fundació La Marató TV3, and La Caixa Foundation. We also thank the Advanced Light Microscopy Unit at the CRG for their assistance and the Cellex Foundation for providing research facilities and equipment. M.A.M.-R. acknowledges funding from the European Research Council under the 7th Framework Program (FP7/2010-2015, ERC grant agreement 609989), the European Union's Horizon 2020 research and innovation program (agreement 676556), the Spanish Ministry of Economy and Competitiveness (BFU2013-47736-P), and the Centro de Excelencia Severo Ochoa 2013-2017 (SEV-2012-0208) to the CRG.

## Author contributions

L.P.-R., V.D.C., A.I., G.S.-B., and J.P.C.-C. performed the experiments. E.B., M.A.M.-R., S.G., and L.N. analyzed the data with contributions from L.D.C., F.L.D., Y.C., and A.G.H. L.P.R. and S.P. designed the experiments and wrote the paper.

## Additional information

**Competing interests:** The authors declare no competing interests.

