## [Peer Review File · Nature Communications]

Reviewers' comments:

Reviewer #1 (Remarks to the Author):

The manuscript describes an analysis of nucleoplasmic lamin B1 protein and its association with euchromatic transcribed regions during the epithelial to mesenchymal transition.

The manuscript is written in a very concise manner and it would benefit from a more detailed description of the background literature as well as the findings. Some of the results in the manuscript are quite different from the established dogma in the field but the authors do not discuss these differences in detail or attempt to reconcile their results with previous findings. The following are more specific concerns:

1. Page 2. Authors mention that "As shown in the bioanalyzer intensity profile, these conditions generated a significant fraction of fragments larger than 1 Kb, which mainly corresponded to heterochromatin. Thus, although heterochromatin was present in our samples after chromatin precipitation, this fraction was size-excluded from Illumina sequencing (Extended Data Fig. 1b)". This seems like an important point that should be explained in more detail. How was it determined that fragments larger than 1 kb correspond to heterochromatin, and that these fragments were excluded from Illumina sequencing? After sequencing, did the authors examine the sizes of inserts and found no inserts in this size range?
2. Figure 1A. It would help the reader understand the significance in the distribution of Lamin B1 peaks if the figure included the location of the genes in this region of the genome.
3. Page 4. Authors conclude that "In summary, LB1 can be found in expressed euchromatin regions associated with C/G regions that are gene-rich, accessible, decorated with euchromatin histone marks and change dynamically during EMT transformation". Authors should explain why LB1 has not been found associated with active genes before. Lamin-associated regions are usually identified using DamID, instead of ChIP. Is there an explanation for why DamID would not find association between LB1 and active genes? In the manuscript by Gesson et al, cited by the authors, LB1 was only found associated with heterochromatin. Authors should discuss these discrepancies between their results and those already published.
4. Page 4. The number of processed reads for Hi-C libraries is quite low and may not allow the authors to achieve the resolutions they claim and to adequately map the changes in compartments and TADs they describe. For example, the authors claim that 50% of TAD borders were conserved during EMT, but if the resolution in mapping these TAD boundaries is 100 kb, it is possible that changes in the location of TAD boundaries within this distance cannot be detected.
5. Authors should consider the possibility that the effect of LB1 at TAD borders is indirect and due to effects on transcription.
6. Authors should discuss in more detail the significance of HP1a findings, which are only described in passing. What is the relevance of HP1a to the rest of the work presented in the manuscript?

Reviewer #2 (Remarks to the Author):

The manuscript by Pasual-Reguant et al. performed lamin B1 ChIP-seq and Hi-C mapping to understand how lamin B1 might regulate chromatin interactions to control TGF-beta-induced EMT. The genomic data presented in the manuscript are useful information to record changes of lamin B1 interaction with chromatin. However, the biology significance of this study is unknown due to the lack of functional data demonstrating whether these lamin 1/chromatin interactions regulate

EMT.

1. No experimental data are provided to support the conclusion in the abstract that " eLADs are required for this cellular transformation". Lamin B1 knockdown experiments were performed to show that reduction of Lamin B1 resulted in reduction of LB1+ sites, changes of gene expressions, reduced EMT marker changes, and reduced migration and invasion. However, given the broad role of Lamin B1 in nuclear organization, it is unclear whether these are due to the eLADs or other functions of Lamin B1. Lentivirus-mediated gene knockdown takes several days to generate stable cell lines, the statement that knockdown of Lamin B1 only affects non-integrated LB1 is uncertain.
2. The entire study is based on using the NMuMG cell line. It is unclear whether all the results presented support a conserved mechanism of TGF-beta-induced EMT or are specific to the experimental system used.
3. It is unclear how lamin B1 is able to specifically bind many genes associated with EMT. No conserved DNA motifs or chromatin characteristics are identified to provide much needed molecular insights into how lamin B1 regulates gene expression. It is possible that lamin B1 has a general role in helping chromatin rearrangement during gene turn ON/OFF in response to various transcription factor-binding events, instead of a unique role in EMT. Although the current study generates large amount of nice genomic data, these data alone do not improve our understanding of lamin B1 regulation of gene expression or TGF-beta-induced EMT.

Response to the reviewers' comments:

Reviewer #1:

The manuscript describes an analysis of nucleoplasmic lamin B1 protein and its association with euchromatic transcribed regions during the epithelial to mesenchymal transition.

The manuscript is written in a very concise manner and it would benefit from a more detailed description of the background literature as well as the findings. Some of the results in the manuscript are quite different from the established dogma in the field but the authors do not discuss these differences in detail or attempt to reconcile their results with previous findings.

We apologize for our overly concise descriptions in the manuscript; we agree with the reviewer that extending this would help make it more understandable. We have now divided the manuscript into subsections and extended each section. This has allowed us to discuss in more detail why our results are in apparent contradiction from the established dogma.

Along this line, it is true that so far no one has been able to map lamin B1 to euchromatin regions; however, negative results are never conclusive. For instance, while the DamID technique is very powerful and useful (Greil *et al.*, 2006), it is known that methylation by Dam proteins is not perfect and that a high level of biases are introduced using this technique: first, the intrinsic affinity of Dam for GATC sequences causes considerable levels of background; and second, methylation levels are also affected by chromatin accessibility and protein turn-over.

To distinguish specific binding from this background signal, it is important to compare the methylation profile obtained with Dam alone (i.e., Dam not fused to a DNA-binding protein). The binding at particular loci is expressed as the ratio of methylation for Dam-fusion:Dam, thereby normalizing the background methylation. If we take this into consideration, together with the fact that there are two different pools of lamin B1 (one in the nuclear envelope, which is more abundant and more stable, and one in the nucleoplasmic fraction, which is less abundant and has a faster turnover), it is then possible that the specific lamin B1 signal in euchromatin is unwittingly discarded after the normalization of the background methylation. We would like to note that while DamID has been used to map DNA binding sites of transcription factors in euchromatin, these proteins do not have the same high level of complexity of localization (with an extra pool in another nuclear localization) that is observed for lamin B1.

Regarding the ChIP-seq experiments: in general, researchers do not enrich samples based on sonication conditions. In fact, this enrichment was only performed in the work of Gesson *et al.* (2016) (which was very inspiring for us). Indeed, many of the lamin A/C regions described in the Gesson *et al.* article overlap with our eLADS (Figure 2), which makes sense, as lamins (A/C and B) form an interconnected network. However, it is true that Gesson *et al.* did not detect enrichment of lamin B1 in euchromatin regions. In this regard, we can speculate (based on our current knowledge) that filaments of lamins vary between cell types. Thus, differences between our results and those presented in Gesson *et al.* could potentially be due to distinct configurations of cell types, and/or to the experimental fact that different antibodies were used in the two studies.

The following are more specific concerns:

1. Page 2. Authors mention that “As shown in the bioanalyzer intensity profile, these conditions generated a significant fraction of fragments larger than 1 Kb, which mainly corresponded to heterochromatin. Thus, although heterochromatin was present in our samples after chromatin precipitation, this fraction was size-excluded from Illumina sequencing (Extended Data Fig. 1b)”. This seems like an important point that should be explain in more detail.

We have extended the degree of explanation in this section. We agree with the reviewer that our experimental settings are a key point in this work, and we would like to thank her/him for bringing up this concern.

How was it determined that fragments larger than 1 kb correspond to heterochromatin, and that these fragments were excluded from Illumina sequencing?

It has been long known that there are genomic regions resistant to sonication, and that these regions are mainly formed by heterochromatin (Auerbach et al., 2009; Becker et al., 2017; Frenster et al., 1963; Gesson et al., 2016; Horvath and Horz, 1981; Mieczkowski et al., 2016). Under low sonication conditions (i.e., with fewer sonication rounds), heterochromatin is not fragmented but rather makes up large fragments that are not transformed into clusters for its further sequencing. (Please also see https://support.illumina.com/sequencing/sequencing_instruments/cluster_station/questions.html).

After sequencing, did the authors examine the sizes of inserts and found no inserts in this size range? As we used single-end reads in our sequencing experiments, we were not able to examine the size of the inserts.

2. Figure 1A. It would help the reader understand the significance in the distribution of Lamin B1 peaks if the figure included the location of the genes in this region of the genome.

We agree; Figure 1 now shows the gene locations.

3. Page 4. Authors conclude that “In summary, LB1 can be found in expressed euchromatin regions associated with C/G regions that are gene-rich, accessible, decorated with euchromatin histone marks and change dynamically during EMT transformation”. Authors should explain why LB1 has not been found associated with active genes before. Lamin-associated regions are usually identified using DamID, instead of CHIP. Is there an explanation for why DamID would not find association between LB1 and active genes? In the manuscript by Gesson et al, cited by the authors, LB1 was only found associated with heterochromatin. Authors should discuss these discrepancies between their results and those already published.

Please see our main answer on the first page, which address this issue.

4. Page 4. The number of processed reads for Hi-C libraries is quite low and may not allow the authors to achieve the resolutions they claim and to adequately map the changes in compartments and TADs they describe. For example, the authors claim that 50% of TAD borders were conserved during EMT, but if the resolution in mapping these TAD boundaries is 100 kb, it is possible that changes in the location of TAD boundaries within this distance cannot be detected.

The reviewer is correct in wondering about whether the amount of produced reads is enough for TADbit to assign TAD borders. Indeed, TADbit (as all the TAD border detection algorithms) has a dependency on the number of reads used to generate the interaction matrices. This has been quantified by the group of Biccato in their recent article in *Nature Methods* (Forcato et al., 2017) (in particular Sup. Fig 16). Importantly, the “accuracy”—or at least the reproducibility (Jaccard Index)—of the TADbit TAD border assignments between replicates at ~100M reads is ~0.4. The results from this independent experiment thus indicate that TADbit is able to reproduce TAD border assignment as well as other methods between replicates at the same level of coverage as the ones introduced here. Nevertheless, border changes within the resolution of the matrices cannot be discarded.

5. Authors should consider the possibility that the effect of LB1 at TAD borders is indirect and due to effects on transcription.

The reviewer is absolutely right. Solely based on the data we presented originally, we were not able to discard that transcription played a role in determining border strength. For this reason, we have now analyzed whether there are changes in transcription in those genes located at the borders at the onset of EMT. We observed that these genes are transcribed throughout EMT with no differences between time points (now shown as Figure 3b). This result makes us believe that contributions of transcription to the strength of the border is minimal if at all.

6. Authors should discuss in more detail the significance of HP1a findings, which are only described in passing. What is the relevance of HP1a to the rest of the work presented in the manuscript?

We have been working with the EMT model for many years now. In addition to the classical migration and invasion assays, there is an easy readout assay that determines the time point during EMT at which HP1 is released from the chromocenters to allow chromatin to be reorganize and to acquire the mesenchymal chromatin organization that we routinely test in our EMT experiments (Millanes-Romero et al., 2013). However, after the reviewer's comment, we think that adding this additional information might be more confusing than helpful and have now removed this section from the revised version of the manuscript.

Reviewer #2:

The manuscript by Pasual-Reguant et al. performed lamin B1 ChIP-seq and Hi-C mapping to understand how lamin B1 might regulate chromatin interactions to control TGF-beta-induced EMT. The genomic data presented in the manuscript are useful information to record changes of lamin B1 interaction with chromatin. However, the biology significance of this study is unknown due to the lack of functional data demonstrating whether these lamin 1/chromatin interactions regulate EMT.

1. No experimental data are provided to support the conclusion in the abstract that “ eLADs are required for this cellular transformation”.

We agree with the reviewer. We cannot conclude that eLADs are required for this cellular transformation. We have softened this sentence to indicating that the levels of lamin B1 in euchromatin (ChIP-PCR in KD conditions showed lamin B1 decrease at euchromatin regions but the amount of lamin B1 in conventional LADs regions were maintained (Fig. 4d and 4e) are critical for EMT.

Lamin B1 knockdown experiments were performed to show that reduction of Lamin B1 resulted in reduction of LB1+ sites, changes of gene expressions, reduced EMT maker changes, and reduced migration and invasion. However, given the broad role of Lamin B1 in nuclear organization, it is unclear whether these are due to the eLADs or other functions of Lamin B1. Lentivirus-mediated gene knockdown takes several days to generate stable cell lines, the statement that knockdown of Lamin B1 only affects non-integrated LB1 is uncertain.

The purpose of the knockdown (KD) experiments was to analyse whether lamin B1 has a functional role. We agree with the reviewer that the broad role of lamin B1 in nuclear organization could have explained the functional impairment that we observed in EMT, and we therefore attempted to reduce this chance by taking the following precautions:

1. Lentivirus-mediated KD was done in transient conditions. All data were obtained 48 hours post-puromycin selection;
2. We choose the shRNA with the mildest effects on lamin B1 levels (as determined by qRT-PCR and western blots);
3. Proliferation assays and nuclear staining showed no differences between control and KD conditions. Strong reductions of LB1 levels are translated into reduction of cell proliferation and entrance into the senescence state. We did not see any of these

phenotypes, and we are still able to detect lamin B1 in conventional LADs by ChIP-PCR, which suggest that we are not affecting the broader, more general role of lamin B1;

4. In this new version of the manuscript, we also demonstrated that this shRNA did not induce senescence, another phenotype associated with strong lamin B1 reduction (Figure 5c).

Still, this is an important question, and we have tried to demonstrate in more detail how our strategy mainly affected the nucleoplasmic pool of lamin B1. In order to answer this question, we did:

1. Subcellular fractionation and western blot (now Fig. 4b): these new data show that the nucleoplasmic lamin B1 (faster turnover) is more affected than the nuclear envelope–lamin B1 fraction.
2. Fluorescence recovery after photo-bleaching (FRAP) experiments also show these differences in the turnover and that, under KD conditions, the nucleoplasmic fraction is the one that is more affected by the shRNA (Fig. 4c)

2. The entire study is based on using the NMuMG cell line. It is unclear whether all the results presented support a conserved mechanism of TGF-beta-induced EMT or are specific to the experimental system used.

The reviewer is right. However, the main objective of this work was to demonstrate not only the existence of eLADs but also their dynamics. Since this objective would require an extensive generation of genomic data (RNA-seq, ATAC-seq, ChIP-seq and HiC at each time point), we decided to choose this system based in our extensive knowledge of this model. Thus, while it is true that, for now, we cannot extend the results obtained here to other systems, we think we are opening a door for identifying these new domains in other models, such as in embryonic stem cell differentiation.

3. It is unclear how lamin B1 is able to specifically bind many genes associated with EMT. No conserved DNA motifs or chromatin characteristics are identified to provide much needed molecular insights into how lamin B1 regulates gene expression.

We apologize if we did not explain ourselves well and would like to clarify this point. In untreated conditions/epithelial states, GO analysis (now shown in Supplementary Figure 2b) shows that lamin B1 enrichment is not biased towards EMT genes. Lamin B1-occupied genes are involved in general transcription and in many different pathways, suggesting that these domains can be related with many pathways. However, it is true that we have not added molecular insight to the lamin B1 reorganization during EMT towards genes involved in EMT. Which proteins help lamin B1 bind euchromatin genes? Which are the molecular mechanisms that govern its redistribution during the EMT? We have analyzed the enrichment motifs in the new LB1 sites that are formed after TGF- β treatment. In Supplementary Figure 2b, we now show that the factors found are strongly related with the TGF- β pathway. Many questions remain to be answered that are beyond the scope of this initial article, which was focused on demonstrating the existence of this yet-unknown chromatin domains in at least this cellular model.

It is possible that lamin B1 has a general role in helping chromatin rearrangement during gene turn ON/OFF in response to various transcription factor-binding events, instead of a unique role in EMT.

We appreciate this comment, which reflects exactly what we meant. We have extended the discussion of the manuscript trying to explain this point better. We also did an experiment for the reviewer that shows that, under lamin B1 KD conditions, neural differentiation of embryonic stem cells (ESCs) is also affected (Figure R1 for reviewer 2). Reduced levels of lamin B1 did not affect cell viability or pluripotency, as has been previously reported (Kim et al., 2011; Kim et al., 2013). Regarding ESC differentiation, previous experiments reported that knocking down lamins still gave normal *in vitro* differentiation, but only a few markers were analyzed in this study (Kim et al., 2013). Moreover, a recent paper demonstrated that lamin B1 is required for the up-regulation of lineage-specific genes,

suggesting a role for lamin B1 in differentiation and gene expression (Gigante et al., 2017), which is in line with our results. This figure illustrates that lamin B1 could be an important player in other cellular processes. Of course, much additional work should be done in the context of ES differentiation, for example by identifying eLADs, carrying out HiC experiments, etc. At the moment, this is beyond the scope of this manuscript, but we believe our results open an interesting line of future investigations. We believe that these data (Fig. R1) should not be included in the manuscript; however, if the reviewer or the editor consider this experiment to be important, we will include it.

Although the current study generates large amount of nice genomic data, these data alone do not improve our understanding of lamin B1 regulation of gene expression or TGF-beta-induced EMT.

We respectfully disagree with this comment. We believe that demonstration of the existence of a large group of uncharacterized lamin B1 domains in contact with genes that are dynamic and functional at the onset of the EMT greatly improves our understanding of gene regulation.

References

- Auerbach, R.K., Euskirchen, G., Rozowsky, J., Lamarre-Vincent, N., Moqtaderi, Z., Lefrancois, P., Struhl, K., Gerstein, M., and Snyder, M. (2009). Mapping accessible chromatin regions using Sono-Seq. *Proc Natl Acad Sci U S A* *106*, 14926-14931.
- Becker, J.S., McCarthy, R.L., Sidoli, S., Donahue, G., Kaeding, K.E., He, Z., Lin, S., Garcia, B.A., and Zaret, K.S. (2017). Genomic and Proteomic Resolution of Heterochromatin and Its Restriction of Alternate Fate Genes. *Molecular cell* *68*, 1023-1037 e1015.
- Forcato, M., Nicoletti, C., Pal, K., Livi, C.M., Ferrari, F., and Bicciato, S. (2017). Comparison of computational methods for Hi-C data analysis. *Nat Methods* *14*, 679-685.
- Frenster, J.H., Allfrey, V.G., and Mirsky, A.E. (1963). Repressed and Active Chromatin Isolated from Interphase Lymphocytes. *Proc Natl Acad Sci U S A* *50*, 1026-1032.
- Gesson, K., Rescheneder, P., Skoruppa, M.P., von Haeseler, A., Dechat, T., and Foisner, R. (2016). A-type lamins bind both hetero- and euchromatin, the latter being regulated by lamina-associated polypeptide 2 alpha. *Genome Res* *26*, 462-473.
- Gigante, C.M., Dibattista, M., Dong, F.N., Zheng, X., Yue, S., Young, S.G., Reiser, J., Zheng, Y., and Zhao, H. (2017). Lamin B1 is required for mature neuron-specific gene expression during olfactory sensory neuron differentiation. *Nature communications* *8*, 15098.
- Greil, F., Moorman, C., and van Steensel, B. (2006). DamID: mapping of in vivo protein-genome interactions using tethered DNA adenine methyltransferase. *Methods Enzymol* *410*, 342-359.
- Horvath, P., and Horz, W. (1981). The compaction of mouse heterochromatin as studied by nuclease digestion. *FEBS Lett* *134*, 25-28.
- Kim, Y., Sharov, A.A., McDole, K., Cheng, M., Hao, H., Fan, C.M., Gaiano, N., Ko, M.S., and Zheng, Y. (2011). Mouse B-type lamins are required for proper organogenesis but not by embryonic stem cells. *Science* *334*, 1706-1710.
- Kim, Y., Zheng, X., and Zheng, Y. (2013). Proliferation and differentiation of mouse embryonic stem cells lacking all lamins. *Cell Res* *23*, 1420-1423.
- Mieczkowski, J., Cook, A., Bowman, S.K., Mueller, B., Alver, B.H., Kundu, S., Deaton, A.M., Urban, J.A., Larschan, E., Park, P.J., et al. (2016). MNase titration reveals differences between nucleosome occupancy and chromatin accessibility. *Nature communications* *7*, 11485.

Millanes-Romero, A., Herranz, N., Perrera, V., Iturbide, A., Loubat-Casanovas, J., Gil, J., Jenuwein, T., Garcia de Herreros, A., and Peiro, S. (2013). Regulation of heterochromatin transcription by Snail1/LOXL2 during epithelial-to-mesenchymal transition. *Molecular cell* 52, 746-757.

a**b****c**
Figure R1

Figure legend:

a) Western blot analysis of LaminB1 in control (shCTRL) and LaminB1-depleted (shLB1) mouse Embryonic Stem cells (mESCs). GAPDH was used as a loading control. b) RT-qPCR analysis of pluripotency markers in proliferating condition (d0) or after 5 days of differentiation (serum-free medium containing N2B27) (d5) of control (shCTRL) and LaminB1-depleted (shLB1) mESCs. Results were normalized to the housekeeping *RplpO*. Error bars represent the Standard Error of the Mean (SEM) of two independent experiments. c) RT-qPCR analysis of neural markers in proliferating condition (d0) or after 5 days of differentiation (serum-free medium containing N2B27) (d5) of control (shCTRL) and LaminB1-depleted (shLB1) mESCs. Results as in (b).

M&M:

Cell Culture, Generation of Stable Cell Lines and Differentiation Assay

E14TG2a mouse Embryonic Stem cells were kept under feeder-free, 20% serum condition as previously described (Morey et al., 2012). Cells were infected with lentivirus produced in 293T to generate stable cell lines expressing shRNA against LaminB1 or a control shRNA. Cells were selected with puromycin (2 µg/ml) for three days. For neuroectodermal specification, LaminB1 knockdown and control cells were plated in serum/LIF condition (100'000 cells/100x20 Nunc dish). On the following day, the medium was changed with serum-free medium supplemented with N2 and B27. Samples were collected after five days of differentiation.

Reviewers' comments:

Reviewer #1 (Remarks to the Author):

I have read the rebuttal letter and the revised version of the manuscript. The authors have addressed all my concerns and the revised version represents an important contribution to the field of nuclear architecture and the contribution of lamins to chromatin organization

Reviewer #2 (Remarks to the Author):

The revised manuscript attempts to address several key points raised in the previous review by softening several major conclusions stated in the original manuscript. While these changes improved the accuracy of the conclusions that could be reached based on the current data, they also significantly reduced the conceptual advance and the biological significance of the study. Below are specific comments.

1. Previous comment "the biology significance of this study is unknown due to the lack of functional data demonstrating whether these lamin 1/chromatin interactions regulate EMT"-----
No additional data are provided to determine whether the interactions between lamin 1 and chromatin (not lamin 1 protein alone) regulates EMT.

2. Previous Comment Point #1: The added data show that knockdown of LB1 didn't cause major reduction of cell proliferation or induce senescence. But the effect of LB1 knockdown on cell migration and invasion could be complex and due to other effects of LB1 unrelated to eLADs and EMT. Without a mutant form of LB1 that is specifically defective in binding to eLADs or binding to nuclear envelope, the current data are not sufficient to support a specific role of LB1 in regulating eLADs during EMT.

3. Previous Comment Point #2: The Authors state that they focus on only one cell line to show the dynamics of LB1 interaction with eLADs. The key issue is that because there is no-functional demonstration on whether such eLADs truly play a role in TGF-beta-induced EMT. What left for the readers is a large number of changes that occurred in this one cell line upon TGF-beta treatment without functional implication. If similar changes are shown in another cell line in response to TGF-beta treatment, the readers can at least conclude that such chromatin changes are correlated with TGF-beta-induced EMT in more than one cell line. The current manuscript contains neither functional data nor additional correlative data to support the conclusion that these eLAD changes are specific and functional in TGF-beta-induced EMT.

4. Previous Comment Point #3: Although few enrichment motifs are listed in the supplementary figure 2b, no further information is provided on how many other motifs are enriched, and these motifs listed are not specific for TGF-beta signaling. More importantly, the authors agree that LB1 is likely to play a general role in helping chromatin rearrangement in response to various transcription factor-binding events, as previous publications indicated. But the current manuscript provide neither mechanistic nor functional data to further our understanding on how TGF-beta specifically utilizes LB1 to regulates EMT vs. other cellular events.

Reviewers' comments:

Reviewer #3 (Remarks to the Author):

The manuscript describes a highly interesting and novel finding that contradicts the current dogma in the field that lamin B1 associates exclusively with heterochromatic genomic regions. The main and surprising findings are:

Lamin B1 localizes also in the nuclear interior during EMT of mouse mammary epithelial cells

Lamin B1 associates with accessible, euchromatic regions of the genome particularly around TSS

Lamin B1 associated genes change during EMT particularly involving TGF β target genes

Lamin B1 associates with TAD borders correlating with border strength.

Lamin B1 KD impairs EMT and correlates with gene expression changes, 50% of DE genes being lamin B1 target genes.

In general this is a nice study with high novelty and of interest to many scientists in the field. However authors should be more careful with some of their conclusions and interpretation of data, as alternative possibilities may exist (see below). Furthermore, some of the experiments have to be explained in more detail in the text to allow readers not experienced in genome-wide analyses following the rationale of the experiments and getting the main points and conclusions more easily.

Specific comments:

Is there a correlation between lamin B1 binding and gene expression during EMT? The manuscript implies that there is a correlation but this is not directly tested (see comments to Fig. 1 and S1 and S2 below)

The causative role of lamin B1 gene association in EMT is over-interpreted. For this conclusion authors also have to show that lamin B + EMT genes are no longer bound by lamin B in KD cells.

The conclusion of lamin B1 localization in the nuclear interior is not fully supported by the data and alternative interpretations may exist (see Fig. S1 and Fig. 4). Using an epi-fluorescence microscope may produce some intranuclear staining due to out of focus signal from the nuclear envelope. Alternatively, nuclear envelope invaginations often appear as intranuclear localization. The authors ignore the fact that unlike lamin A, lamin B is farnesylated and carboxymethylated and thus tightly attached to the nuclear membrane. Further controls would be needed to support the conclusion that lamin B1 is in the nuclear interior, such as immunostaining of an integrated inner nuclear membrane protein (e.g. LBR that should not be present in the nuclear interior), or solubility of lamin B1 in nuclear fractionation assays using detergent-free buffers rather than chromatin association assays (Fig. 4b). Overall the data are also consistent with a peripheral pool of lamin B1, which is less well integrated into the lamina filament network and thus more dynamic, but still membrane bound. It cannot be excluded that e-LADs are still bound to regions (patches) of the NE, which are either more dynamic and/or are located in invaginations that reach into nucleoplasm. This alternative model predicts association of a group of active genes with the nuclear envelope, which is equally exciting in my opinion.

Line 38: LADs were originally defined as Lamina-associated domains, not lamin-associated domains.

Lane 74: Based on the above definition of LADs, it is incorrect to say "a new set of LADs" referring to lamin-associated domains in the nuclear interior.

Fig. S2: Legend and/or text have to be more precise and explained in more detail. I am puzzled by the presented marks. These are different in the different conditions and H3K27me3 is a repressive mark, yet authors show upregulation of genes in Fig. 1e.

Fig. 1b: Lamin B1 association around promoter regions is unchanged during EMT, although authors

claim lamin B1 associates dynamically with euchromatin during EMT. Are different genes occupied by lamin B1 at the different stages? It would be more interesting and informative to check differentially expressed genes or lamin B+ only genes in treated cells.

Fig. 1 g and h: What do numbers in Venn diagrams mean? If these are genes it is inconsistent with the main text (lane 112, 28% of genes maintain lamin B1). It would also be clearer to adjust circle sizes according to the number of genes.

Fig. 2a. What is different in Figures 1a and 2a and between Figs. 1f and 2b, and what is the rationale of showing these Figures? eLADs could easily be added to Fig. 1a. I understand that the authors analyzed MACS peaks first followed by determination of eLADs. However, since eLADs are derived from MACS Peak distribution I would not expect any difference in the properties of peaks versus cLADs.

Fig. 3b: Lamin B1 association increases with TAD borders during EMT. How does this relate to lamin B1 association with gene promoters?

Fig. 4c: Images are not convincing. Intranuclear lamins are not detectable at all.

Fig. 5 shows control experiments and could be moved to supplement.

Reviewer #2 previous comments to authors:

The revised manuscript attempts to address several key points raised in the previous review by softening several major conclusions stated in the original manuscript. While these changes improved the accuracy of the conclusions that could be reached based on the current data, they also significantly reduced the conceptual advance and the biological significance of the study. Below are specific comments.

1. Previous comment "the biology significance of this study is unknown due to the lack of functional data demonstrating whether these lamin 1/chromatin interactions regulate EMT"-----
No additional data are provided to determine whether the interactions between lamin 1 and chromatin (not lamin 1 protein alone) regulates EMT.

2. Previous Comment Point #1: The added data show that knockdown of LB1 didn't cause major reduction of cell proliferation or induce senescence. But the effect of LB1 knockdown on cell migration and invasion could be complex and due to other effects of LB1 unrelated to eLADs and EMT. Without a mutant form of LB1 that is specifically defective in binding to eLADs or binding to nuclear envelope, the current data are not sufficient to support a specific role of LB1 in regulating eLADs during EMT.

3. Previous Comment Point #2: The Authors state that they focus on only one cell line to show the dynamics of LB1 interaction with eLADs. The key issue is that because there is no-functional demonstration on whether such eLADs truly play a role in TGF-beta-induced EMT. What left for the readers is a large number of changes that occurred in this one cell line upon TGF-beta treatment without functional implication. If similar changes are shown in another cell line in response to TGF-beta treatment, the readers can at least conclude that such chromatin changes are correlated with TGF-beta-induced EMT in more than one cell line. The current manuscript contains neither functional data nor additional correlative data to support the conclusion that these eLAD changes are specific and functional in TGF-beta-induced EMT.

4. Previous Comment Point #3: Although few enrichment motifs are listed in the supplementary figure 2b, no further information is provided on how many other motifs are enriched, and these motifs listed are not specific for TGF-beta signaling. More importantly, the authors agree that LB1

is likely to play a general role in helping chromatin rearrangement in response to various transcription factor-binding events, as previous publications indicated. But the current manuscript provide neither mechanistic nor functional data to further our understanding on how TGF-beta specifically utilizes LB1 to regulates EMT vs. other cellular events.

Response to the reviewers' comments:

Reviewer #3:

In general this is a nice study with high novelty and of interest to many scientists in the field. However authors should be more careful with some of their conclusions and interpretation of data, as alternative possibilities may exist (see below). Furthermore, some of the experiments have to be explained in more detail in the text to allow readers not experienced in genome-wide analyses following the rational of the experiments and getting the main points and conclusions more easily.

We would like to thank the reviewer for her/his constructive comments. We agree that alternative possibilities exist regarding lamin B1 localization, and we now included this possibility in the manuscript. Moreover, we did co-localization immunofluorescence experiments and detergent-free subcellular fractionation, as suggested, to obtain more evidence for the interpretation of our results.

Following the reviewer's suggestion, we have extended the analysis for the genome-wide data. We have also included more details in the text and figure legends, to help the reader understand the main points and conclusions easily.

Specific comments:

Is there a correlation between lamin B1 binding and gene expression during EMT? The manuscript implies that there is a correlation but this is not directly tested (see comments to Fig. 1 and S1 and S2 below)

We thank the reviewer for this interesting observation. Indeed, when we integrate our RNA-seq expression profiles during EMT into our current lamin B1 ChIP-seq genomic landscape, it is immediately evident that most expression peaks precisely overlap our eLADs regions. To analyse this issue in detail, we have conducted several bioinformatics experiments:

- (i) First, we stratified the full set of mouse genes into four groups according to their RNA-seq expression, of silent, low, medium, or high (for each time point separately). Next, we generated the meta-gene plot of the corresponding lamin B1 ChIP-seq sample around the TSS of each group of genes. In the three time points (untreated, 8 h, and 24 h), we were able to distinguish the characteristic ChIP binding signal pattern of each of the gene sets grouped by expression. In general, the higher the expression, the higher the strength of the ChIP binding.
- (ii) Second, we performed the test the other way around (only untreated is shown, we obtained equivalent results for all time points). As for (i), we divided all genes in the genome into the four groups according to the max peak of lamin B1 ChIP-seq in their promoter. Next, we evaluated the expression level at 0 h in the RNA-seq experiment to confirm whether the difference in binding could be reflected in the expression of the same genes. Indeed, by analysing the distribution of gene expression in a boxplot, a clear and distinct pattern of expression was revealed for each group of genes: the higher the ChIP signal, the higher the expression of the gene marked by lamin B1.

(iii) Finally, we directly calculated the correlation between RNA-seq RPKMs and ChIP-seq normalized reads of lamin B1 for all genes in the untreated condition (as well as for 8 h and 24 h; the results were equivalent). The correlation coefficient between both features was 0.44 (positive correlation).

- We did not include these last analyses (ii and iii) into the manuscript, as we believe that they provide redundant information; rather, we prepared a figure for the reviewer (Figure R1), which could be included into the manuscript as part of a main figure or as a supplementary figure if the reviewer deems necessary.

To conclude from all these results, we consider that there is a correlation between lamin B1 binding and gene expression during EMT. We have included (i) in the revised version of the manuscript (Fig. 1g).

The causative role of lamin B1 gene association in EMT is over-interpreted. For this conclusion authors also have to show that lamin B + EMT genes are no longer bound by lamin B in KD cells.

This interpretation was made based in several pieces of evidence:

1. An increased presence of lamin B1 in genes involved in EMT after induction of EMT;
2. An altered transcription profile at 8 h upon TGF β that is lamin B1 dependent. More than 50% of the genes are direct lamin B1 targets;
3. Changes in the EMT-transcriptional program when lamin B1 is knocked down
4. Impairment of EMT in lamin B1 KD conditions
5. We have added now new genome-wide analysis showing that there is a correlation between lamin B1 binding and gene expression during EMT.

However, we agree with the reviewer that without an experiment showing that lamin B1 no longer binds EMT genes in KD conditions, the causative role maybe over-interpreted. We did qChIP-PCR for a subset of selected EMT genes under control and KD conditions following TGF β treatment. The new Figure 6f shows that lamin B1 binding is lost in KD in selected EMT genes at 8 h after TGF β treatment.

Nonetheless, we have tried to soften our conclusions to avoid over-interpretation.

The conclusion of lamin B1 localization in the nuclear interior is not fully supported by the data and alternative interpretations may exist (see Fig. S1 and Fig. 4). Using an epi-fluorescence microscope may produce some intranuclear staining due to out of focus signal from the nuclear envelope.

We agree; however, we would like to clarify that all microscopy analysis shown in the manuscript was done using confocal rather than epi-fluorescence microscopy. We have added this detail in the text and in the figure legend and apologize for any confusion.

Alternatively, nuclear envelope invaginations often appear as intranuclear localization. The authors ignore the fact that unlike lamin A, lamin B is farnesylated and carboxymethylated and thus tightly

attached to the nuclear membrane. Further controls would be needed to support the conclusion that lamin B1 is in the nuclear interior, such as immunostaining of an integrated inner nuclear membrane protein (e.g. LBR that should not be present in the nuclear interior), or solubility of lamin B1 in nuclear fractionation assays using detergent-free buffers rather than chromatin association assays (Fig. 4b). Overall the data are also consistent with a peripheral pool of lamin B1, which is less well integrated into the lamina filament network and thus more dynamic, but still membrane bound. It cannot be excluded that e-LADs are still bound to regions (patches) of the NE, which are either more dynamic and/or are located in invaginations that reach into nucleoplasm. This alternative model predicts association of a group of active genes with the nuclear envelope, which is equally exciting in my opinion.

We appreciate this comment. We have now added to the introduction and discussion (and have referenced the sources) the fact that lamin B1 is farnesylated and carboxymethylated and therefore tightly attached to the nuclear membrane. We also appreciated the proposal of an alternative interpretation in which lamin B1 would be less integrated into the lamina filament, making it more dynamic. This interpretation also fits our data on lamin B1 stability and euchromatin binding, and we also find it to be equally exciting.

Nevertheless, following the reviewer's suggestion, we did co-localization experiments using confocal microscopy and subcellular fractionation in the absence of detergent. We did a co-localization experiment between lamin B1 and emerlin, a conserved LEM-domain protein, in NMUMG cells following by fluorescence signal quantification. As the best LBR antibodies have been removed from the market, we choose emerlin as an integral membrane protein that localizes at the nuclear envelope (Berk et al., 2013). We still observed lamin B1 staining in the nucleoplasm that did not overlap with emerlin (Supplementary Fig. 1)

Detergent-free subcellular fractionation was also done as the reviewer suggested, and we also were able to detect lamin B1 in the nucleoplasmic fraction (Fig. R2).

After discussing this in-depth, we believe (despite these results) that we cannot discard the existence of lamin B1 patches located in invaginations that reach the nucleoplasm; we have now tried to make this clear throughout the manuscript and in the discussion section.

Line 38: LADs were originally defined as Lamina-associated domains, not lamin-associated domains. We apologize for this mistake, which has now been corrected.

Lane 74: Based on the above definition of LADs, it is incorrect to say "a new set of LADs" referring to lamin-associated domains in the nuclear interior.

If the reviewer agrees, we can change it for eLADs, to differentiate from LADs that are associated with heterochromatin.

Fig. S2: Legend and/or text have to be more precise and explained in more detail. I am puzzled by the presented marks. These are different in the different conditions and H3K27me3 is a repressive mark, yet authors show upregulation of genes in Fig. 1e.

We apologize for the incomplete labelling of our previous Figure S2a. In order to clearly present this

data, we have reformulated the results by summarizing all the histone marks that appear as significant ($p < 0.001$) at each time point, rather than only showing the top 3. Thus, with this new representation of the same information, we believe that the reader can appropriately examine which marks appear/disappear on the lamin B1 target genes during EMT.

In particular, we have determined that a mixture of euchromatin histone marks as well as H3K27me3 (a repressive mark associated with Polycomb) are associated with genes positive for lamin B1. We think that it is not surprising, as many developmental genes are only active in particular cell lines/tissues during development and are silent in the rest of the organism.

In order to elucidate the canonical signature of heterochromatic regions (which should be different from that of euchromatin, and therefore should not contain H3K27me3), we repeated the same analysis on the genes that do not belong to our eLADs at each point (they are not targeted by lamin B1). Here, we did see the characteristic enrichment on the H3K9me3 mark (which was not reported previously in our set of lamin B1 genes).

The Enrichr tool takes all the published data into account to provide a probability. From this, we can say that our list of genes is strongly associated with euchromatin histone marks in different situations but never with heterochromatin histone marks, further supporting the fact that lamin B1+ genes are located in euchromatin regions rather than in heterochromatin. We have now improved the explanation about this in the manuscript.

Fig. 1b: Lamin B1 association around promoter regions is unchanged during EMT, although authors claim lamin B1 associates dynamically with euchromatin during EMT. Are different genes occupied by lamin B1 at the different stages? It would be more interesting and informative to check differentially expressed genes or lamin B+ only genes in treated cells.

A Venn diagram shows the different lamin B1+ genes during the EMT. Enrichment is maintained in the TSS, but the specific genes differ during the process.

Following the reviewer's suggestions, we have now included differentially expressed genes and lamin B1+ in treated cells. Two new Venn diagrams (Fig. 2a) show these new results.

Fig. 1 g and h: What do numbers in Venn diagrams mean? If these are genes it is inconsistent with the main text (line 112, 28% of genes maintain lamin B1). It would also be clearer to adjust circle sizes according to the number of genes.

We apologize if this panel was not clear enough. We showed the number of genes that were enriched in lamin B1 (line 148–151). Here, we can see the dynamism of the enrichment (e.g., loss of lamin B1 in some genes, and gain of lamin B1 in others).

Fig. 2a. What is different in Figures 1a and 2a and between Figs. 1f and 2b, and what is the rationale of showing these Figures? eLADs could easily be added to Fig. 1a. I understand that the authors analyzed MACS peaks first followed by determination of eLADs. However, since eLADs are derived from MACS Peak distribution I would not expect any difference in the properties of peaks versus cLADs.

We agree with the reviewer that, for the eLADs in Figure 3, we are reproducing some of the analyses shown previously in Figure 1 for the ChIP-seq peaks of lamin B1. However, we consider that these assessments are necessary to confirm that we observed the same properties in the eLADs as observed previously for ChIP-seq peaks. It should be taken into account that, to define an eLAD, we first must identify two peaks of the ChIP-seq at a maximum distance with no other genes (inside this region) that is not marked by lamin B1. Therefore, the fact that we identified the same features in eLADs that were reported before for peaks is remarkable. In addition, we defined eLADs to follow the canonical view in the literature that such binding events constitute regions rather than isolated peaks. Thus, we were able to compare our sets of eLADs with those published for lamin A/C in a previous publication and to evaluate the genome coverage of each catalogue of sites. However, to provide a new information as compared to Figure 1, we have now incorporated our RNA-seq profiles into Figure 3, to emphasize for the reader the strong overlap between eLADs and clusters of expressed genes in this genomic region.

Fig. 3b: Lamin B1 association increases with TAD borders during EMT. How does this relate to lamin B1 association with gene promoters?

Promoters of active genes are known to be enriched in TAD borders (Dixon et al., 2012; Ea et al., 2015), which is in agreement with the increased number of genes with lamin B1 enrichment we observed following TGF β treatment. We have extended the text in order to clarify this point.

Fig. 4c: Images are not convincing. Intranuclear lamins are not detectable at all.

We included the quantification analysis ($n = 10$) due to the rather low intensity of lamin B1 fused-cerulean, figure brightness has been increased also. One reason, however, why the reviewer cannot observe the intranuclear signal that we observe could be the file transformation (from .tiff to .pdf) or the screen computer resolution, as we have observed the same signal using different computer monitors.

Fig. 5 shows control experiments and could be moved to supplement.

We followed the reviewer suggestion; these results are shown now as Supplementary Figure. 5

Reviewer #2:

Reviewer #2 previous comments to authors:

The revised manuscript attempts to address several key points raised in the previous review by softening several major conclusions stated in the original manuscript. While these changes improved the accuracy of the conclusions that could be reached based on the current data, they also significantly reduced the conceptual advance and the biological significance of the study. Below are specific comments.

We really do not understand how “the improvement of the accuracy of the conclusions reduces the conceptual advance and the biological significance of the study”.

1. Previous comment “the biology significance of this study is unknown due to the lack of functional data demonstrating whether these lamin 1/chromatin interactions regulate EMT”----- No additional data are provided to determine whether the interactions between lamin 1 and chromatin (not lamin 1 protein alone) regulates EMT.

- a. We show that lamin B1 that contacts euchromatin is reduced under lamin B1 knock-down conditions by qChIP-PCR. Concomitantly, there is an alteration of all the EMT transcriptional program, and cells do not behaved as mesenchymal cells upon TGF- β induction;
- b. Upon TGF- β induction, we show that new lamin B1 sites contacting euchromatin are formed in genomic regions that are directly related with the TGF- β pathway and the EMT process;
- c. We have shown that lamin B1 knockdown mainly affects the nucleoplasmic fraction (in which almost all euchromatin was located). Moreover, we have shown that nucleoplasmic lamin B1 reduction did not cause either changes in gene expression or alterations of cell survival, apoptosis, or proliferation. Indeed, changes that we observed were in migration and invasion as well as in the classical markers of the EMT process;
- d. If we understand correctly, we believe that he/she is suggesting more functional experiments to determine if the interactions between lamin B1 and chromatin modulates (or regulates) EMT without affecting the lamin B1 protein alone. We believe that the only way to do this would be to using a mutant, as he/she suggested again later (please see our response in the following comment).

2. Previous Comment Point #1: The added data show that knockdown of lamin B1 didn't cause major reduction of cell proliferation or induce senescence. But the effect of lamin B1 knockdown on cell migration and invasion could be complex and due to other effects of lamin B1 unrelated to eLADs and EMT. Without a mutant form of lamin B1 that is specifically defective in binding to eLADs or binding to nuclear envelope, the current data are not sufficient to support a specific role of lamin B1 in regulating eLADs during EMT.

Reviewer #2 suggests that we could use a lamin B1 mutant to show that it binds euchromatin but not to chromatin that is associated with the nuclear envelope. While the structure of lamins in the nuclear envelope has recently been determined (Turgay et al, Nature 2017), nothing is known yet about its structure in the nucleoplasmic fraction. Thus, there is no way to generate a mutant that only affects lamin B1 binding to a particular chromatin fraction.

But the effect of lamin B1 knockdown on cell migration and invasion could be complex and due to other effects of lamin B1 unrelated to eLADs and EMT

We find hard to envision a scenario in which the effects of depleting lamin B1 from euchromatin would affect EMT behaviour in an EMT-unrelated way

3. Previous Comment Point #2: The Authors state that they focus on only one cell line to show the

dynamics of lamin B1 interaction with eLADs. The key issue is that because there is no-functional demonstration on whether such eLADs truly play a role in TGF-beta-induced EMT. What left for the readers is a large number of changes that occurred in this one cell line upon TGF-beta treatment without functional implication. If similar changes are shown in another cell line in response to TGF-beta treatment, the readers can at least conclude that such chromatin changes are correlated with TGF-beta-induced EMT in more than one cell line. The current manuscript contains neither functional data nor additional correlative data to support the conclusion that these eLAD changes are specific and functional in TGF-beta-induced EMT.

We would like to emphasize the fact that we chose the EMT system based on our extensive previous experience with it and to further demonstrate that eLADs are functional in a physiologically-relevant cell transformation context—but that we believe that this mechanism (of eLAD regulation) is not restricted to the EMT process. Maybe this was not clear enough for reviewer #2, despite our attempt to explain this in the first revision; we apologize if so. We also included experimental data in embryonic stem cells and during differentiation in our previous revision. Further, we stressed in the manuscript that our results are not meant to preclude the existence of eLADs and their functionality in other cell types and cellular transformations; in fact, it is quite the opposite (we believe that this will be a major mechanism of regulation in distinct cell types/processes). In other words, eLADs are not likely to be EMT exclusive; we hope that we were able to now make our point evident for reviewer #2.

4. Previous Comment Point #3: Although few enrichment motifs are listed in the supplementary figure 2b, no further information is provided on how many other motifs are enriched, and these motifs listed are not specific for TGF-beta signaling. More importantly, the authors agree that lamin B1 is likely to play a general role in helping chromatin rearrangement in response to various transcription factor-binding events, as previous publications indicated. But the current manuscript provide neither mechanistic nor functional data to further our understanding on how TGF-beta specifically utilizes lamin B1 to regulates EMT vs. other cellular events.

We showed significantly enriched domains (a *p*-value is used for the ranking). We would like to clarify that some motifs are indeed part of the TGF-beta signaling pathway, either directly (SMAD4) or indirectly (EGR1). Some references include:

1. EGR1 (<https://www.uniprot.org/uniprot/P08046>): a transcription factor that activates TGFβ (Forte et al., 2017; Krones-Herzig et al., 2005)
2. SMAD4 (<https://www.uniprot.org/uniprot/P97471>). The classical co-activator of the TGFβ pathway.

The rest of motifs are related with other transformation processes, such as muscle differentiation (MYOD), hematopoietic development (MZF1), or differentiation processes (GATA3), which is in agreement with our proposal involving transcription factor-dependent processes and lamin B1-dependent chromatin reorganization.

We have included now the full list of transcription factors that we identified (as Supplementary Table 1) together with a GSEA to show how all the transcription factors we identified are involve in developmental programs.

Berk, J. M., Tifft, K. E., and Wilson, K. L. (2013). The nuclear envelope LEM-domain protein emerin. *Nucleus* 4, 298-314.

Dixon, J. R., Selvaraj, S., Yue, F., Kim, A., Li, Y., Shen, Y., Hu, M., Liu, J. S., and Ren, B. (2012). Topological domains in mammalian genomes identified by analysis of chromatin interactions. *Nature* 485, 376-380.

Ea, V., Baudement, M. O., Lesne, A., and Forne, T. (2015). Contribution of Topological Domains and Loop Formation to 3D Chromatin Organization. *Genes (Basel)* 6, 734-750.

Forte, E., Chimenti, I., Rosa, P., Angelini, F., Pagano, F., Calogero, A., Giacomello, A., and Messina, E. (2017). EMT/MET at the Crossroad of Stemness, Regeneration and Oncogenesis: The Ying-Yang Equilibrium Recapitulated in Cell Spheroids. *Cancers (Basel)* 9.

Krones-Herzig, A., Mittal, S., Yule, K., Liang, H., English, C., Urcis, R., Soni, T., Adamson, E. D., and Mercola, D. (2005). Early growth response 1 acts as a tumor suppressor in vivo and in vitro via regulation of p53. *Cancer Res* 65, 5133-5143.

REVIEWERS' COMMENTS:

Reviewer #3 (Remarks to the Author):

The revised manuscript has significantly improved and addresses the most important concerns I had in my previous evaluation. Importantly, the authors included additional bioinformatics analyses showing a positive correlation of lamin B1 binding to genes at TSS and gene expression during EMT. In addition, they soften their previous conclusions on lamin B1 localization in the nuclear interior and mention alternative interpretations of the results.

Overall, this manuscript describes convincingly that laminB1 binds to open chromatin (euchromatin) on upregulated genes during EMT in NMuMG cells. This is a surprising and unexpected finding in view of the prevailing opinion that lamin B1 exclusively binds heterochromatic regions at the nuclear envelope. However, there is increasing evidence from several labs using different methodologies and cell systems that various components of the peripheral lamina interact not only with silenced heterochromatin but also with highly expressed genes in euchromatin. The manuscript by Pascual-Reguant et al. supports this emerging new concept and adds lamin B1 to the list of nuclear lamina proteins that have much more complex roles in chromatin organization than previously thought.

Response to the reviewers' comments:

Reviewer #3:

The revised manuscript has significantly improved and addresses the most important concerns I had in my previous evaluation. Importantly, the authors included additional bioinformatics analyses showing a positive correlation of lamin B1 binding to genes at TSS and gene expression during EMT. In addition, they soften their previous conclusions on lamin B1 localization in the nuclear interior and mention alternative interpretations of the results.

Overall, this manuscript describes convincingly that laminB1 binds to open chromatin (euchromatin) on upregulated genes during EMT in NMuMG cells. This is a surprising and unexpected finding in view of the prevailing opinion that lamin B1 exclusively binds heterochromatic regions at the nuclear envelope. However, there is increasing evidence from several labs using different methodologies and cell systems that various components of the peripheral lamina interact not only with silenced heterochromatin but also with highly expressed genes in euchromatin. The manuscript by Pascual-Reguant et al. supports this emerging new concept and adds lamin B1 to the list of nuclear lamina proteins that have much more complex roles in chromatin organization than previously thought.

We would like to thank the reviewer for all her/his constructive comments. We also appreciated the proposal of an alternative interpretation in which lamin B1 would be less integrated into the lamina filament, making it more dynamic. This interpretation also fits our data on lamin B1 stability and euchromatin binding, and we also find it to be equally exiting.